# Topological Data Analysis on Graphs: Euler characteristics, Persistent Homology, or Spectrum?

## Abstract

Graph neural networks (GNNs) are limited by the Weisfeiler-Leman (WL) hierarchy and cannot compute graph properties such as cycles. Topological descriptors (TDs) such as the Euler characteristics (EC), persistent homology (PH), and Laplacian spectrums have thus been employed to enhance the GNNs. However, despite empirical successes, the theoretical underpinnings of these TDs remain largely underexplored. We bridge this gap with a rigorous characterization of TDs focusing on three key aspects: expressivity (representational power), stability (robustness to data perturbations), and computation (implementation cost). We evaluate the expressivity of different TDs, and design a novel scheme $\text{RePHINE}^{Spec}$ that is strictly more expressive. We also propose new metrics to assess the stability of the state-of-the-art RePHINE method and the newly proposed $\text{RePHINE}^{Spec}$ method. To address computational costs, we introduce and analyze weaker variants for several descriptors. TDs find significant applications in molecular contexts, so we also explore new filtration functions on the molecular graphs. Finally, we formalize the properties of filtration functions derived from graph products. Overall, this work lays the foundation for the principled design and analysis of new TDs that can be tailored to specific applications.

## 1 Introduction

Message-passing GNNs are prominent models for graph representation learning (Gilmer et al., 2017). However, they are bounded in expressivity by the WL hierarchy (Xu et al., 2019; Morris et al., 2019; Maron et al., 2019; Morris et al., 2024) and cannot compute fundamental graph properties such as cycles or connected components (Garg et al., 2020; Chen et al., 2020). TDs such as those based on PH can provide such information and thus are being increasingly employed to augment GNNs, boosting their empirical performance (Carriere et al., 2020; Zhao et al., 2020; Chen et al., 2021; Horn et al., 2022). However, barring a few notable exceptions such as Immonen et al. (2023); Ballester & Rieck (2024), the expressivity analysis of TDs remain rather elusive.

In this work, we study three prominent types of topological descriptors based on PH (Edelsbrunner & Harer, 2008), EC (Turner et al., 2014; Röell & Rieck, 2024), and the Laplacian spectrum (Wang et al., 2020; Bakó et al., 2022) respectively. On a high level, PH keeps track of the birth and death times of topological features (e.g., connected components and loops) in a (parameterized) graph filtration. EC is a simpler, less expensive invariant that keeps track of the number of vertices minus the number of edges in the filtration. The Laplacian spectrum keeps track of the eigenvalues of the graph Laplacian through the filtration, which neither PH nor EC can account for.

Not much is known about the expressivity of TDs. Recently, Immonen et al. (2023) analyzed PH under color filtrations, providing a complete characterization of 0-dimensional PH in vertex-level and edge-level filtrations using graph-theoretic notions; and introduced *RePHINE* as a strictly more powerful scheme than both. However, the expressivity of EC and spectrum TDs remains unexplored, motivating our first set of investigations: *how do different topological descriptors stack up against each other, and can their strengths be unified to design even more expressive TDs than RePHINE?*

Here, we offer a complete characterization for the expressivity of EC in terms of combinatorial data on the colors of the graph. In particular, it turns out that EC is strictly less expressive than PH, ruling

out any further efforts to combine the two for representational benefits. On the other hand, Laplacian appears in several flavors: unlike graph Laplacian (the 0-th combinatorial Laplacian), rows in the 1-st combinatorial Laplacian correspond to the edges of the graph (as opposed to vertices), and persistent versions (Wang et al., 2020) have also been proposed for both. We show that, surprisingly, the expressivity offered by the non-zero eigenvalues of all these spectral TDs can simply be replicated by including the non-zero eigenvalues of graph Laplacians at all time steps (Appendix B). Armed with this key insight, we introduce a new topological scheme called RePHINE$^{Spec}$ that amalgamates RePHINE and graph Laplacians to be strictly more expressive than all the existing TDs.

However, expressivity is not the only design consideration: in general, topological features are expensive to compute. Unlike expressivity, EC compares favorably with other TDs in terms of computation. This leads us to our next phase of analyses: *can we design computationally cheaper variants of TDs*? We thus proceed to streamline RePHINE, RePHINE$^{Spec}$, and EC with what we call their "max" versions. We show that max EC is even more cost-effective than EC, while still retaining the latter's expressivity. Interestingly, we show that this expressivity equivalence on graphs extends to a more general equivalence for color filtrations on finite higher-dimensional simplicial complexes (Appendix C). This immediately opens avenues for integrating max EC as an economical alternative to PH-based TDs into topological neural networks, which generalize GNNs via higher order message-passing (Papillon et al., 2024; Papamarkou et al., 2024; Eitan et al., 2024) and have recently been shown to benefit from TDs both theoretically and empirically (Verma et al., 2024).

Besides expressivity and computation, stability is another important issue. A diagram-based TD is *stable* if a small perturbation in filtration incurs only a small change in the diagram it produces. Stability has been used, among others, to compare and classify geometric shapes (Cohen-Steiner et al., 2006). Stability results are known for usual persistence diagrams in Cohen-Steiner et al. (2006) and EC in Dłotko & Gurnari (2023); however, even defining a suitable metric to quantify stability is challenging for RePHINE since it strictly generalizes PH with node-colors, but all known metrics for the persistence diagrams in edge-level filtrations are node-color agnostic. We fill this gap by motivating a novel metric for measuring stability and proving that RePHINE is globally stable under this metric. We also prove RePHINE$^{Spec}$ is locally stable under a similar metric.

From a practical micro-level perspective, the design of effective filtration functions is also paramount. Typically, filtration functions are either fixed *a priori* or learned without any structural considerations, obfuscating how they are about the corresponding diagrams. Instead, we initiate here a formal analysis by characterizing filtrations on graph products: we establish that the product of vertex filtrations is the max of vertex colors, and that the product of edge filtrations is the natural edge coloring on the product. We also provide an algorithm to track the evolution of 0-dimensional persistence diagram under the product filtration induced by edge filtrations.

*Isomorphism invariance* is another desiderata required to ensure that isomorphic graphs produce identical diagrams. While PH and RePHINE were already shown to be isomorphic invariant (Ballester & Rieck, 2024; Immonen et al., 2023), we establish the same for all the other aforementioned TDs.

Finally, since TDs have been found to be particularly useful in the context of molecular data (Horn et al., 2022; Immonen et al., 2023; Verma et al., 2024), we investigate whether there can be any further benefits to viewing the edges (pertaining to the bonds between the nodes, i.e., atoms) as topological objects themselves in place of their usual color-based representation. We establish that there are no differences between the two on 0-th dimensional persistence diagrams or on EC in edge-level filtrations, and they are incomparable in 1-dimensional diagrams.

In sum, we build a firm foundation for principled design and analysis of TDs. We summarize our key contributions in Figure 1, and relegate all the proofs to Appendix A.

## 2 PRELIMINARIES AND SETUP

Unless mentioned otherwise, we will be considering graphs $G = (V, E, c, X)$ with finite vertex set $V$, edges $E \subseteq V \times V$, and a vertex-coloring function $c : V \to X$, where $X$ is a finite set denoting the space of available colors or features. All graphs are simple unless mentioned otherwise. Two graphs $G = (V, E, c, X)$ and $G' = (V', E', c', X')$ are isomorphic if there is a bijection $h : V \to V'$

| **Main contributions of this work** | |
| --- | --- |
| **Expressive Power of Filtration Methods (Section 3, Appendix B, C)** | |
| Construction of Spectral RePHINE Diagrams | Definition 6 |
| Graph of Expressivity Comparisons | Theorem 1 |
| EC Diagram $\cong$ max EC Diagram | Theorem 2 |
| The Expressive Power of EC Diagrams on Graphs | Theorem 3 |
| Sufficiency of Graph Laplacian in $\mathrm{RePHINE}^{Spec}$ | Corollary 2 |
| The Expressive Power of EC Diagrams on Simplicial Complexes | Theorem 6 |
| **Stability of RePHINE and $\mathrm{RePHINE}^{Spec}$ Diagrams (Section 4):** | |
| Construction of a suitable metric $d_B^R$ on RePHINE | Definition 7 |
| RePHINE is (globally) stable under $d_B^R$ | Theorem 4 |
| Construction of a suitable metric $d_B^{\mathrm{Spec}\,R}$ on $\mathrm{RePHINE}^{Spec}$ | Definition 8 |
| $\mathrm{RePHINE}^{Spec}$ is locally stable under $d_B^{\mathrm{Spec}\,R}$ | Theorem 5 |
| **Filtrations on Graph Products (Section 5):** | |
| Product of Vertex Filtrations $\simeq$ Vertex Coloring on Product | Proposition 3 |
| Product of Edge Filtrations $\simeq$ Edge Coloring on Product | Proposition 4 |
| Algorithm to Compute 0-th Persistence Pairs for Edge Filtrations | Proposition 5 |
| **Well-Definedness of Filtration Methods (Section D):** | |
| $\mathrm{RePHINE}^{Spec}$ (and more) are Isomorphism Invariant | Proposition 19 |
| The Inconsistency of Death Time Filtrations | Example 2 |
| **Filtrations on Molecular Graphs (Section 6):** | |
| Equivalence of 0-th Dimensional Persistence Diagram | Proposition 6 |
| Equivalence of EC in Edge Filtrations | Proposition 7 |

Figure 1: Overview of our results.

of the vertices such that (1) the two coloring functions are related by $c = c' \circ h$ and (2) the edge $(v, w)$ is in $E$ if and only if $(h(v), h(w))$ is in $E'$.

We remark the first condition ensures that isomorphic graphs should share the same coloring set. For example, the graph $K_3$ with all vertices colored "red" will not be isomorphic to the graph $K_3$ with all vertices colored "blue", as it fails the first condition. For the rest of this work, we will assume that any two graphs $G$ and $H$ **share the same coloring set** $X$ (we can always without loss take $X$ to be the union of their coloring sets). Since we will never talk about more than two graphs at a time, we will also assume that **all graphs that appear have the coloring set** $X$.

**Definition 1** (Coloring Filtrations). *On a color set $X$, we consider the pair of functions $(f_v : X \to \mathbb{R}, f_e : X \times X \to \mathbb{R}_{>0})$ where $f_e$ is symmetric (ie. $f_e(c, c') = f_e(c', c)$). On a graph $G$ with a vertex color set $X$, $(f_v, f_e)$ induces the following pair of functions $(F_v, F_e)$.*

*1. For all $v \in V(G)$, $F_v(v) := f_v(c(v))$. For all $e \in E(G)$ with vertices $v_1, v_2$, $F_v(e) = \max\{F_v(v_1), F_v(v_2)\}$. Intuitively, we are assigning the edge $e$ with the color $c(\arg\max_{v_i} F_v(v_i))$ (the vertex color that has a higher value under $f_v$).*

*2. For all $v \in V(G)$, $F_e(v) = 0$. For all $e \in e(G)$ with vertices $v_1, v_2$, $F_e(e) := f_e(c(v_1), c(v_2))$. Intuitively, we are assigning the edge $e$ with the color $(c(v_1), c(v_2))$.*

*For each $t \in \mathbb{R}$, we write $G_t^{f_v} := F_v^{-1}((-\infty, t])$ and $G_t^{f_e} := F_e^{-1}((-\infty, t])$.*

Note we used $G_t^{f_v}$ as opposed to $G_t^{F_v}$ to emphasize that the function $G \mapsto (\{G_t^{f_v}\}_{t \in \mathbb{R}}, \{G_t^{f_e}\}_{\mathbb{R}})$ is well-defined for any graph $G$ with the coloring set $X$. The lists $\{G_t^{f_v}\}_{t \in \mathbb{R}}$ and $\{G_t^{f_e}\}_{t \in \mathbb{R}}$ define a **vertex filtration** of $G$ by $F_v$ and an **edge filtration** of $G$ by $F_e$ respectively. It is clear that $G_t^{f_v}$ can

only change when $t$ crosses a critical value in $\{f_v(c) : c \in X\}$, and $G_t^{f_e}$ can only change when $t$ crosses a critical value in $\{f_e(c_1, c_2) : (c_1, c_2) \in X \times X\}$. Hence, we can reduce both filtrations to finite filtrations at those critical values.

We now review three prominent classes of topological descriptors (TDs), namely, persistent homology, Euler characteristic, and Laplacian spectrum. All methods we define in this section are persistent versions of these TDs, that is, we want to keep track of how they evolve over time.

## 2.1 Persistent Homology

A vertex $v$ (ie. $0$-dimensional persistence information) is born when it appears in a given filtration of a diagram. When we merge two connected components represented by two vertices $v$ and $w$, we use a decision rule to kill off one of the vertices and mark the remaining vertex to represent the new connected component. A cycle (ie. $1$-dimensional persistence information) is born when it appears in a given filtration of a diagram, and it will never die. For color-based vertex and edge filtration, there is a canonical way to calculate the **persistence pairs** of a graph with a given filtration. We refer the reader to Appendix A of Immonen et al. (2023) for a precise introduction. We say a $0$-th dimensional persistence pair $(b, d)$ is a **real hole** if $d = \infty$, is an **almost hole** if $b \neq d < \infty$, and is a **trivial hole** if $b = d$. Note that edge-based filtrations do not have any trivial holes.

**Definition 2.** *Let $f = (f_v, f_e)$ be on the coloring set $X$. The persistent homology (PH) diagram of a graph $G$ is a collection $\mathrm{PH}(G, f)$ composed of two lists $\mathrm{PH}(G, f)^0, \mathrm{PH}(G, f)^1$ where $\mathrm{PH}(G, f)^0$ are all the persistent pairs in the vertex filtration $\{G_t^{f_v}\}_{t \in \mathbb{R}}$ and $\mathrm{PH}(G, f)^1$ are all the persistent pairs in the edge filtration $\{G_t^{f_e}\}_{t \in \mathbb{R}}$.*

PH is an isomorphism invariant (Theorem 2 of Ballester & Rieck (2024)) and can be used in GNNs. However, PH alone does not account for sufficient local color information. Consequently, Immonen et al. (2023) introduced the RePHINE diagram as a generalization of PH.

**Definition 3.** *Let $f = (f_v, f_e)$ be on $X$. The **RePHINE diagram** of a graph $G$ is a multi-set $\mathrm{RePHINE}(G, f) = \mathrm{RePHINE}(G, f)^0 \sqcup \mathrm{RePHINE}(G, f)^1$ of cardinality $|V(G)| + \beta_G^1$ where:*

- *0-th dimensional component: $\mathrm{RePHINE}(G, f)^0$ consists of tuples of the form $(b(v), d(v), \alpha(v), \gamma(v))$ for each vertex $v \in V(G)$. Here, $b(v)$ and $d(v)$ are the birth and death times of $v$ under the edge filtration $\{G_t^{f_e}\}_{t \in \mathbb{R}}$, $\alpha(v) = f_v(c(v))$ and $\gamma(v) = \min_{v \in N(w)} f_e(c(v), c(w))$. Here, $N(\omega)$ denotes the neighboring vertices of $\omega$.*

  *The decision rule for which vertex to kill off is as follows - an almost hole $(b, d)$ corresponds to the merging of two connected components with vertex representatives $v_1$ and $v_2$. We kill off the vertex that has a greater value under $\alpha$. If there is a tie, we kill off the vertex that has a lower value under $\gamma$. If there is a further tie, Theorem 4 of Immonen et al. (2023) shows that the resulting diagram $\mathrm{RePHINE}(G, f)^0$ is independent of which choice we make here.*

- *1-st dimensional component: $\mathrm{RePHINE}(G, f)^1$ consists of tuples of the form $(1, d, 0, 0)$ where each $d$ indicates the birth time of a cycle in the same filtration. In the definition of Immonen et al. (2023), the birth of a cycle corresponds to what is so-called the death of a "missing hole". This is why we use $d$ to indicate the birth time instead.*

Theorem 5 of Immonen et al. (2023) asserts that RePHINE diagrams are strictly more expressive than PH diagrams. Here, we introduce a new variant of the RePHINE diagram with an added constraint on $f_e$.

**Definition 4.** *Let $f_v : X \to \mathbb{R}$ be a vertex function such that $f_v > 0$. The **max RePHINE diagram** of a graph $G$ is a multi-set $\mathrm{RePHINE}^m(G, f_v) = \mathrm{RePHINE}(G, f_v, f_e)$ where $f_e(c_1, c_2) = \max(f_v(c_1), f_v(c_2))$. Intuitively, for each edge $e = (v_1, v_2) \in E(G)$, we assign $e$ the color value of whichever vertex $v_i$ that has a greater value under $F_v$.*

Note that Definition 4 is a special case of Definition 3. The setting of max RePHINE is more suitable for certain physics-informed systems where the vertex might have a strong influence on the attributes of its neighboring edges. Another advantage is that the max RePHINE diagram only needs to keep track of the vertex colors as it assigns each edge a color in $X$. This is in contrast to the more general RePHINE diagram which assigns each edge a color in $X \times X$. The usage of max RePHINE uses less memory in the space of possible edge colors than the RePHINE diagram.

## 2.2 EULER CHARACTERISTIC

Including topological features in graph representation learning using persistent homology or RePHINE diagrams can be computationally expensive. The Euler characteristic (EC) provides a weaker isomorphism invariant that is easier to compute (but typically less expressive). For a graph $G$, the Euler characteristic $\chi(G)$ is given by $\chi(G) = \#V(G) - \#E(G)$. Given a filtration of a graph $G$, we can track its Euler characteristic throughout the filtration.

**Definition 5.** *Let $f = (f_v, f_e)$ be on the coloring set $X$. Write $a_1 < ... < a_n$ as the list of values $f_v$ can produce, and $b_1 < ... < b_m$ as the list of values $f_e$ can produce. The **EC diagram** of a graph $G$ is two lists $\mathrm{EC}(G, f) = \mathrm{EC}(G, f)^0 \sqcup \mathrm{EC}(G, f)^1$, where $\mathrm{EC}(G, f)^0$ is the list $\{\chi(G_{a_i}^{f_v})\}_{i=1}^n$ and $\mathrm{EC}(G, f)^1$ is the list $\{\chi(G_{b_i}^{f_e})\}_{i=1}^m$. In the specific case where the pair $f = (f_v, f_e)$ are given as in Definition 4, we define the **max EC diagram** of $G$ as $\mathrm{EC}^m(G, f_v) := \mathrm{EC}(G, f)$.*

## 2.3 LAPLACIAN SPECTRUM

Homology captures harmonic information (in the sense that the kernel of the graph Laplacian corresponds to the 0-th homology), but there is some non-harmonic information we also want to account for. One option is to account for colors, as we have done with RePHINE. Another option is to augment RePHINE with spectral information. We therefore propose a new descriptor below.

**Definition 6.** *Let $f = (f_v, f_e)$ be on the coloring set $X$. The **spectral RePHINE diagram** of a graph $G$ is a multi-set $\mathrm{RePHINE}^{Spec}(G, f) = \mathrm{RePHINE}^{Spec}(G, f)^0 \sqcup \mathrm{RePHINE}^{Spec}(G, f)^1$ of cardinality $|V(G)| + \beta_G^1$ where:*

- *0-th dimensional component: $\mathrm{RePHINE}^{Spec}(G, f)^0$ consists of tuples of the form $(b(v), d(v), \alpha(v), \gamma(v), \rho(v))$ for each vertex $v \in V(G)$. Here, $b, d, \alpha, \gamma$ are the same as Definition 3, and $\rho(v)$ is the list of non-zero eigenvalues of the graph Laplacian of the connected component $v$ is in when it dies at time $d(v)$.*

- *1-st dimensional component: $\mathrm{RePHINE}^{Spec}(G, f)^1$ consists of tuples of the form $(1, d(e), 0, 0, \rho(e))$. Here, $d(e)$ indicates the birth time of a cycle given by the edge $e$. $\rho(e)$ denotes the non-zero eigenvalues the graph Laplacian of the connected component $e$ is in when it is born at time $d(e)$.*

*In the specific case where the pair $f = (f_v, f_e)$ are given as in Definition 4, we also define the **max spectral RePHINE diagram** of $G$ as $\mathrm{RePHINE}^{mSpec}(G, f_v) := \mathrm{RePHINE}^{Spec}(G, f)$.*

An example that computes all the diagrams defined in this section can be found in Figure 2. In the construction of the max versions, our choice of $f_e$ was based on the birth time of vertices under $f_v$. In Appendix D, we conjure a "death-time filtration" as dual to the "birth-time" filtrations and discuss some of its implications.

# 3 THE EXPRESSIVE POWER OF FILTRATION METHODS

## 3.1 COMPARISON OF EXPRESSIVITY

Let $X, Y$ be two graph isomorphism invariants. We say $X$ has **at least the same expressivity** as $Y$ (denoted $X \succeq Y$) if for all non-isomorphic graphs $G$ and $H$ that $Y$ can tell apart, $X$ can also tell them apart. We say $X$ is **strictly more expressive** than $Y$ (denoted $X \succ Y$) if, in addition, there exist two non-isomorphic graphs $G$ and $H$ that $Y$ cannot tell apart but $X$ can. We say $X$ and $Y$ have the **same expressive power** (denoted $X = Y$) if $X \succeq Y$ and $Y \succeq X$. We say $X$ and $Y$ are **incomparable** if there are two non-isomorphic graphs $G$ and $H$ that $X$ can tell apart but $Y$ cannot, and vice versa.

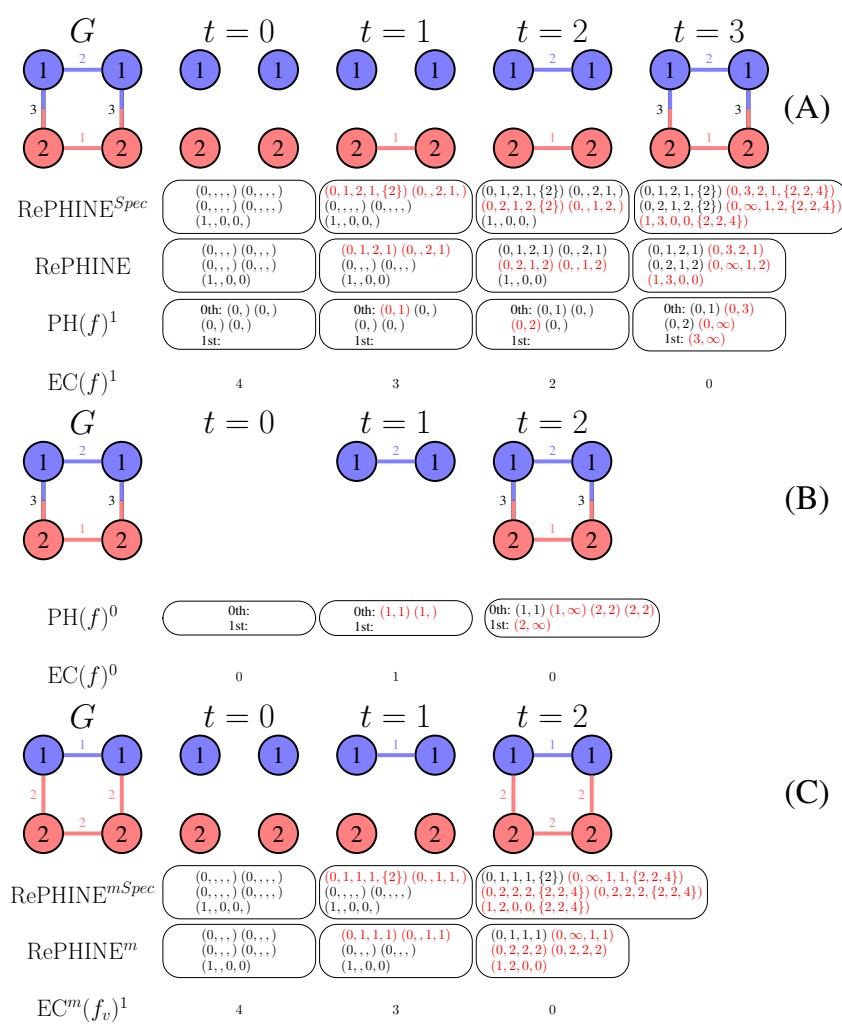

Figure 2: Example computing all the diagrams in Section 2 on a graph $G$ with $f_v(\text{blue}) = 1$, $f_v(\text{red}) = 2$. (A) illustrates an edge filtration of $G$ by $f_e(\text{red}) = 1$, $f_e(\text{blue}) = 2$, $f_e(\text{red-blue}) = 3$. (B) illustrates a vertex filtration of $G$ by $f_v$. (C) illustrates an edge filtration of $G$ induced by $f_v$ in the setting of max diagrams (See Definition 4).

**Theorem 1.** *We have the following graph comparing the expressivity of the diagrams defined in Section 2. Here, a directed arrow $X \to Y$ indicates that $X \succ Y$.*

$$\text{RePHINE}^{Spec} \longrightarrow \text{RePHINE} \longrightarrow \text{PH} \longrightarrow \text{EC}$$

$$\text{RePHINE}^{mSpec} \longrightarrow \text{RePHINE}^{m} \longrightarrow \text{EC}^{m}$$

*incomparable*

Theorem 1 shows that $\text{RePHINE}^{Spec}$ is a strictly more powerful method than all other methods outlined in Section 2. Despite its power, there are still graphs that $\text{RePHINE}^{Spec}$ cannot tell apart (see Figure 3(c)). Figure 3(a) and Figure 3(b) also illustrate with explicit examples that $\text{RePHINE}^{mSpec}$ and RePHINE are incomparable in terms of expressivity. Theorem 1 also implies the following surprising equivalence between max EC diagrams and EC diagrams on graphs.

**Theorem 2.** EC *and* $\text{EC}^{m}$ *have the same expressive power.*

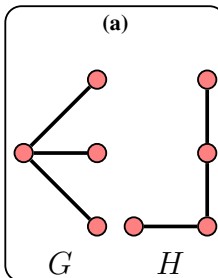 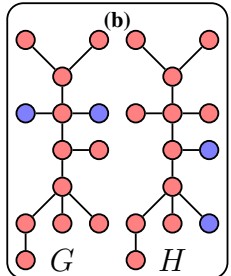 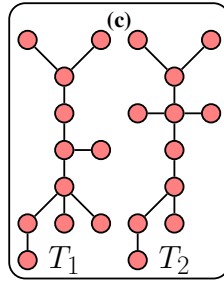

Figure 3: (a) Graphs that $\mathrm{RePHINE}^{Spec}$ and $\mathrm{RePHINE}^{mSpec}$ can tell apart but RePHINE cannot. (b) Graphs that $\mathrm{RePHINE}^{Spec}$ and RePHINE can tell apart but $\mathrm{RePHINE}^{mSpec}$ cannot. (c) Graphs that $\mathrm{RePHINE}^{Spec}$ cannot tell apart.

Unlike the other comparison results in Theorem 1, Theorem 2 will follow as an immediate corollary of a complete characterization of the expressivity of EC diagrams on graphs.

### 3.2 EXPRESSIVITY OF EC ON GRAPHS AND HIGHER SIMPLICIAL COMPLEXES

In this subsection, we will first obtain two complete characterizations of the expressivity of EC with respect to vertex filtrations and edge filtrations separately. Let $G, H$ be two graphs on $n$ vertices.

**Notation:** We use $E_G(a, b)$ to denote the set of edges in $G$ with endpoints being a vertex of color $a$ and a vertex of color $b$, $V_G(a)$ to denote the set of vertices in $G$ with color $a$, and $G(c), H(c)$ to denote the subgraphs of $G, H$ generated by the vertices of color $c$.

In our next results, Proposition 1 and Proposition 2, we establish that the (max) EC diagrams of a graph $G$ may be completely interpreted in terms of the combinatorial data provided by the objects $E_G(a, b), V_G(a)$ and $G(c)$ defined in the notations above.

**Proposition 1** (Characterization of (max) EC Edge Filtrations). *The following are equivalent:*

1. *For all (symmetric) edge color functions $g : X \times X \to \mathbb{R}$, $\mathrm{EC}(G, g)^1 = \mathrm{EC}(H, g)^1$.*
2. *For all vertex color functions $f : X \to \mathbb{R}$, $\mathrm{EC}^m(G, f)^1 = \mathrm{EC}^m(H, f)^1$.*
3. *$\#E_G(a, b) = \#E_H(a, b)$ for all colors $a, b \in X$.*

**Proposition 2** (Characterization of (max) EC Vertex Filtrations). *The following are equivalent:*

1. *For all vertex color functions $f : X \to \mathbb{R}$, $\mathrm{EC}(G, f)^0 = \mathrm{EC}(H, f)^0$.*
2. *For all vertex color functions $f : X \to \mathbb{R}$, $\mathrm{EC}^m(G, f)^0 = \mathrm{EC}^m(H, f)^0$.*
3. *For all $a \neq b, c \in X$, $\chi(G(c)) = \chi(H(c))$ and $\#E_G(a, b) = \#E_H(a, b)$.*

Combining the statements of Proposition 1 and Proposition 2, we obtain the following theorem.

**Theorem 3** (Characterization of (max) EC Diagrams). *The following are equivalent:*
1. *$G$ and $H$ have the same EC diagram for any choice of coloring functions.*
2. *$G$ and $H$ have the same max EC diagram for any choice of coloring functions.*
3. *$\#V_G(c) = \#V_H(c)$ for all $c \in X$ and $\#E_G(a, b) = \#E_H(a, b)$ for any $a, b \in X$.*

Theorem 3 gives a complete characterization for the expressivity of EC and max EC. The equivalence of (1) and (2) in particular proves Theorem 2.

While the majority of our work is focused on graphs, we remark that the equivalence of EC and max EC is a special case of a more general equivalence between the two that occurs in the color-based filtrations of a finite simplicial complex (see Appendix C).

## 4 THE STABILITY OF REPHINE AND $\mathrm{RePHINE}^{Spec}$ DIAGRAMS

Let $f = (f_v : X \to \mathbb{R}, f_e : X \times X \to \mathbb{R}_{>0})$ and $g = (g_v : X \to \mathbb{R}, g_e : X \times X \to \mathbb{R}_{>0})$ be two pairs of functions on $X$. For a graph $G$ with the coloring set $X$, we would ideally like a

way to measure how much the diagrams we constructed in Section 2 differ with respect to $f$ and $g$. One way to measure this is to impose a suitable metric on the space of diagrams and obtain a stable bound. For PH diagrams on $G$, this suitable metric is called the **bottleneck distance** and has been classically shown to be bounded by $||f_v - g_v||_\infty + ||f_e - g_e||_\infty$ (Cohen-Steiner et al., 2006). In this section, we will discuss a generalization of this result to RePHINE and RePHINE$^{Spec}$ diagrams.

Let us examine RePHINE first. One naive proposal for a suitable "metric" on RePHINE diagrams would be to restrict only to the first two components ($b$ and $d$) of the multi-set and use the classical bottleneck distance for persistence diagrams. However, this approach ignores any information from the $\alpha$ and $\gamma$ components and will fail the non-degeneracy axioms for a metric. Thus, we need to modify the metric on RePHINE diagrams to take into account of its $\alpha$ and $\gamma$ components.

**Definition 7.** *Let* $\mathrm{RePHINE}(G, f)$ *and* $\mathrm{RePHINE}(G, g)$ *be the two associated RePHINE diagrams for G respectively. We define the **bottleneck distance** as* $d_B^R(\mathrm{RePHINE}(G, f), \mathrm{RePHINE}(G, g)) :=$
$d_B^{R,0}(\mathrm{RePHINE}(G, f)^0, \mathrm{RePHINE}(G, g)^0) + d_B^{R,1}(\mathrm{RePHINE}(G, f)^1, \mathrm{RePHINE}(G, g)^1).$

*Here,* $d_B^{R,0}$ *(resp.*$d_B^{R,1}$*) is the infimum of distances for which there is a bijection between the two* $\mathrm{RePHINE}^0$ *(resp.*$\mathrm{RePHINE}^1$*) diagrams. The distance is given by*

$$d((b_0, d_0, \alpha_0, \gamma_0), (b_1, d_1, \alpha_1, \gamma_1)) = \max\{|b_1 - b_0|, |d_1 - d_0|\} + |\alpha_1 - \alpha_0| + |\gamma_1 - \gamma_0|.$$

Similarly, we define a metric between $\mathrm{RePHINE}^{Spec}$ diagrams as follows.

**Definition 8.** *We define the **bottleneck distance** between* $\mathrm{RePHINE}^{Spec}$ *diagrams as*

$$d_B^{\mathrm{Spec}\,R}(\mathrm{RePHINE}^{Spec}(G, f), \mathrm{RePHINE}^{Spec}(G, g)) :=$$

$$d_B^{\mathrm{Spec}\,R,0}(\mathrm{RePHINE}^{Spec}(G, f)^0, \mathrm{RePHINE}^{Spec}(G, g)^0)$$

$$+ d_B^{\mathrm{Spec}\,R,1}(\mathrm{RePHINE}^{Spec}(G, f)^1, \mathrm{RePHINE}^{Spec}(G, g)^1)$$

*Here,* $d_B^{\mathrm{Spec}\,R,0}$ *(resp.* $d_B^{\mathrm{Spec}\,R,0}$*) is the infimum of distances for which there is a bijection between the two diagrams. The distance is given by*

$$d'((b_0, d_0, \alpha_0, \gamma_0, \rho_0), (b_1, d_1, \alpha_1, \gamma_1, \rho_1)) = d((b_0, d_0, \alpha_0, \gamma_0), (b_1, d_1, \alpha_1, \gamma_1)) + d^{Spec}(\rho_0, \rho_1),$$

*where $d$ is given in Definition 7 and $d^{Spec}$ is given by embedding $\gamma_0$ and $\gamma_1$ as sorted lists (followed by zeroes) into $\ell^1(\mathbb{N})$ and taking their $\ell^1$-distance in $\ell^1(\mathbb{N})$.*

We verify in Appendix A.3 that Definition 7 and Definition 8 are indeed metrics. We also prove that the bottleneck distances of two RePHINE diagrams may be explicitly bounded in terms of the $\ell^\infty$ norms of the input functions, and hence RePHINE diagrams are stable in the following sense.

> **Theorem 4.** $d_B^R(\mathrm{RePHINE}(G, f), \mathrm{RePHINE}(G, g)) \leq 3||f_e - g_e||_\infty + ||f_v - g_v||_\infty$

As a corollary of Theorem 4, we also obtain an explicit bound on the max RePHINE diagrams.

**Corollary 1.** $d_B^R(\mathrm{RePHINE}^m(G, f_v), \mathrm{RePHINE}^m(G, g_v)) \leq 4||f_v - g_v||_\infty.$

RePHINE diagrams are regarded as *globally stable* in the sense that no matter what $f$ and $g$ we choose, their respective RePHINE diagrams are bounded by a suitable norm on $f$ and $g$. Spectral RePHINE diagrams, in contrast, only satisfy a local form of stability. We make precise what *local* means by introducing a suitable topology on the possible space of filtration functions.

After fixing a canonical ordering on $X$ and $X \times X/ \sim$ separately, we may view $f_v$ (resp. $f_e$) as an element of $\mathbb{R}^{n_v}$ (resp. $(\mathbb{R}_{>0})^{n_e}$). Furthermore, if $f_v$ (resp. $f_e$) is injective, it may viewed as an element in $\mathrm{Conf}_{n_v}(\mathbb{R})$ (resp. $\mathrm{Conf}_{n_e}(\mathbb{R}_{>0})$), where $\mathrm{Conf}_{n_v}(\mathbb{R})$ (resp. $\mathrm{Conf}_{n_e}(\mathbb{R}_{>0})$) is the subspace of $\mathbb{R}^{n_v}$ (resp. $\mathbb{R}^{n_e}$) composing of points whose coordinates have no repeated entries. From here we obtain the following theorem.

> **Theorem 5.** *If $f_v$ and $f_e$ are injective, then $f = (f_v, f_e)$ is locally stable on $\mathrm{Conf}_{n_v}(\mathbb{R}) \times \mathrm{Conf}_{n_e}(\mathbb{R}_{>0})$ under $d_B^{\mathrm{Spec}\,R}$. That is, over a graph $G$ with the coloring set $X$, we have:*
>
> $$d_B^{\mathrm{Spec}\,R}(\mathrm{RePHINE}^{Spec}(G, f), \mathrm{RePHINE}^{Spec}(G, g)) \leq 3||f_e - g_e||_\infty + ||f_v - g_v||_\infty,$$
>
> *for all $g = (g_v, g_e)$ sufficiently close to $f$ in $\mathrm{Conf}_{n_v}(\mathbb{R}) \times \mathrm{Conf}_{n_e}(\mathbb{R}_{>0})$.*

Note that the injectivity assumption is necessary, and $\text{RePHINE}^{Spec}$ is not globally stable in general (see Example 1).

## 5 PRODUCT OF GRAPHS

When dealing with large or complex graphs that have the structural property of being some product, it is often easier to work with the components of the graph product rather than the graph as a whole. From a machine learning perspective, filtration functions are also typically learned without any structural considerations, and we are thus motivated to investigate product as a special kind of structure. Given graphs $G$ and $H$, we seek to investigate what kinds of information about the graph product $G \square H$ can be recovered from analyzing filtrations on $G$ and $H$ alone.

**Definition 9.** *Let $G, H$ be graphs, the box product (Cartesian product) of $G$ and $H$ is the graph $G \square H$, where the vertex set of $G \square H$ is the set $\{(g, h) \mid g \in V(G), h \in V(H)\}$ and the edge set is constructed as follows. For vertices $(g_1, h_1)$ and $(g_2, h_2)$, we draw an edge if (1) $g_1 = g_2$ and $h_1 \sim h_2$ in $H$ or (2) $h_1 = h_2$ and $g_1 \sim g_2$ in $G$. Here, by $h_1 \sim h_2$, we mean that $h_1$ and $h_2$ are related by an edge in $H$ (and similarly for $g_1 \sim g_2$). Note that if one of $G$ and $H$ is empty, then the box product is empty.*

Note that we have not assigned a coloring on $G \square H$ yet. Intuitively, we want to assign a convenient coloring so that we can decompose filtration of $G \square H$ into filtrations on $G$ and $H$ respectively. Formally, we want to be considering filtrations of the following form.

**Definition 10.** *Let $G$ and $H$ be graphs with filtrations functions $f_G$ and $f_H$ respectively. We can define a **product filtration** of $G \square H$ with respect to the two filtration functions as the following - let $t \in \mathbb{R}$, then the subgraph $(G \square H)_t$ is exactly $G_t^{f_G} \square H_t^{f_H}$.*

From here, we obtain the following two propositions that relate the product filtrations of $G \square H$ to vertex and edge filtrations of $G$ and $H$ respectively.

**Proposition 3.** *Suppose $f_G$ and $f_H$ are injective vertex color functions whose images do not overlap. The product filtration with respect to $f_G$ and $f_H$ is equivalent to the filtration on $G \square H$ given by the vertex coloring function $F$, where*

$$F((g, h)) = \max(f_G(c(g)), f_H(c(h))), \text{ for all } (g, h) \in V(G \square H).$$

*In particular, this implies that the persistence diagrams of this product filtration are well-defined.*

**Proposition 4.** *Suppose $f_G > 0$ and $f_H > 0$ are injective edge color functions whose images do not overlap. The product filtration with respect to $f_G$ and $f_H$ is equivalent to the filtration on $G \square H$ given by the function $F$, constructed as follows:*

- *$F$ sends all vertices to $0$.*
- *Let $e \in E(G \square H)$ be an edge with vertices $(g_1, h_1)$ and $(g_2, h_2)$. If $g_1 = g_2$, $F(e) = f_H(c(h_1), c(h_2))$. If $h_1 = h_2$, $F(e) = f_G(c(g_1), c(g_2))$.*

*Note that $g_1 = g_2$ and $h_1 = h_2$ cannot both be satisfied on a simple graph. In particular, $F$ is permutation equivariant, so the persistence diagrams of this product filtration are well-defined.*

Following Proposition 4, we can in fact obtain an algorithmic procedure to keep track of how the $0$-th dimensional persistence diagram of $G \square H$ changes under the product of edge filtrations.

---

**Proposition 5.** *Continuing the set-up of Proposition 4, for each $t > 0$, let $g_t^b$ (resp. $h_t^b$) be the number of vertices still alive in $G_t$ (resp. $H_t$) at time $t$ and $g_t^d$ (resp. $h_t^d$) be the number of vertices in $G_t$ (resp. $H_t$) that died on time $t$. The $0$-dimensional persistence diagram of $(G \square H)_t = G_t \square H_t$ with respect to $f$ is described as follows:*

1. *All vertices are born at time $0$.*
2. *Let $t$ be a time where $G_t$ changes (ie. $t$ is a critical value), then the number of vertices that will die at time $t$ is exactly $h_t^b g_t^d$.*
3. *Let $t$ be a time where $H_t$ changes, then the number of vertices that will die at time $t$ is exactly $g_t^b h_t^d$.*

---

## 6    COLORING AND COUNTING REPRESENTATIONS FOR MOLECULES

Let $M$ be a molecule. The standard representation of $M$ as a graph, which we call hereafter as the **Bond Coloring Representation (BColor)**, views the atoms of $M$ as nodes; and adds an edge of color "triple bond", "double bond", or "single bond" to indicate covalent bonds between the atoms. We also consider here a different representation, which we call **Bond Counting Representation (BCount)**: the nodes denote the atoms as before; however, we add 3 edges to denote a triple bond, 2 edges to denote a double bond, and 1 edge to denote a single bond.

BColor is the conventional choice to represent a molecule (Hoogeboom et al., 2022; Xu et al., 2022; Verma et al., 2022; Xu et al., 2023; Song et al., 2024). A molecule with this topological representation (BCount) is not a simple graph - it is rather a graph with multiple edges between the same vertex. This is an example of what is called a delta complex (Section 2.1 of Hatcher (2002)). In contrast, the counting representation is much less common (Kvasnič̌k & Pospíchal, 1989).

One may consider using BCount since it might be able to capture more accurately the strength of the underlying electrostatic interactions. We seek to investigate whether this alternate topological representation would allow topological methods to better differentiate non-isomorphic molecules. For a molecule $M$, we use $M(c), M(t)$ to denote the BColor representation and BCount representation of $M$ respectively. The coloring set of a vertex filtration of $M(c)$ vs. $M(t)$ will be the type of all atoms, and the coloring set of an edge filtration of $M(c)$ vs. $M(t)$ will be the bond type.

**Proposition 6.** *For any vertex color filtration $f_v$ of $M(c)$ and $M(t)$, $f_v$ will produce the same 0-dimensional persistence diagram. For any edge color filtration $f_e$ of $M(c)$ and $M(t)$, $f_e$ will also produce the same 0-th dimensional persistence diagram.*

**Proposition 7.** *Let $M, N$ be molecules. The EC diagram of $M(t)$ and $N(t)$ with respect to all possible $f_e$ are the same if and only if the EC diagram of $M(c)$ and $N(c)$ with respect to all possible $f_e$ are the same.*

We remark that there are examples of graphs $M$, $N$ whose vertex filtration can be told apart by EC with BCount but not BColor (e.g., consider $M$ with 2 atoms of same type $A$ and a double bond between them; and $N$ with 2 atoms of $A$ and a single bond between them). Conversely, there exist examples that can be separated by EC with BColor but not BCount (e.g., consider $M$ with 3 $A$ atoms and 2 single bonds; and $N$ with 3 $A$ atoms but only 1 double bond). This means that the two representations are incomparable under vertex filtrations with EC as well as with 1-dimensional persistence diagrams (using Proposition 6).

## 7    CONCLUSION

We unraveled the theoretical underpinnings of topological descriptors associated with EC, PH, and the Laplacian spectrum focusing on expressivity, stability, and computation. For expressivity, we amalgamated spectral features with RePHINE to craft a strictly more expressive scheme RePHINE$^{Spec}$. For stability, we constructed a notion of bottleneck distance on RePHINE and RePHINE$^{Spec}$. From here, we showed the former is globally stable, and the latter is locally stable. For computation, we introduced and analyzed several new schemes, including a more economical, but equally expressive, variant of EC. We also examined these methods in the context of graph products and molecules. In particular, our results for molecules imply that we can provably augment expressivity of existing color-based schemes with the counting-based features, opening fascinating avenues for applications in synthesis chemistry and drug discovery.

Our work focused on color-based filtrations, and it may be interesting to study these descriptors under other types of filtrations (e.g., based on the degree of vertices). Exploring suitable metrics for assessing the stability of RePHINE$^{Spec}$ is another interesting direction.

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

## A  Proofs

### A.1  Proof of Theorem 1

We will prove Theorem 1 edge by edge in the following sequence of propositions. Note that Theorem 5 of Immonen et al. (2023) already showed that RePHINE is strictly more expressive than PH. We also defer the proof that $\mathrm{EC} = \mathrm{EC}^m$ to Appendix A.2.

**Proposition 8.** $\mathrm{RePHINE}^{Spec}$ *is strictly more expressive than* RePHINE.

*Proof.* From Definition 6, we know that $\mathrm{RePHINE}^{Spec}$ by definition includes all the data RePHINE can have. To see that it is strictly more expressive, we consider the graphs $G$ as the star graph on 4 vertices and $H$ as the path graph on 4 vertices (see Figure 3(a)). Furthermore, we require that every vertex be colored red. From the remarks below Theorem 5 of Immonen et al. (2023), we know that RePHINE cannot differentiate $G$ and $H$. However, the data about the real holes of the respective $\mathrm{RePHINE}^{Spec}$ diagrams of $G$ and $H$ will be different. This is because the eigenvalues of $\Delta_0(G)$ are $\{0, 1, 1, 4\}$ and the eigenvalues of $\Delta_0(H)$ are $\{0, 2, 2 - \sqrt{2}, 2 + \sqrt{2}\}$. $\square$

**Proposition 9.** $\mathrm{RePHINE}^{mSpec}$ *is strictly more expressive than* $\mathrm{RePHINE}^m$.

*Proof.* From Definition 4 and Definiton 6, we know that $\mathrm{RePHINE}^{mSpec}$ by construction has at least the same expressive power as $\mathrm{RePHINE}^m$. To see that $\mathrm{RePHINE}^{mSpec}$ is strictly more expressive, we can use the same example in Proposition 8 since there are no essential differences between $\mathrm{RePHINE}^{Spec}$ and $\mathrm{RePHINE}^{mSpec}$ on graphs whose vertices all have the same color. $\square$

**Proposition 10.** $\mathrm{RePHINE}^{mSpec}$ *and* RePHINE *are incomparable.*

*Proof.* The same example in Proposition 8 provides a pair of graphs that the max spectral RePHINE diagram can tell apart but the RePHINE diagrams cannot. For the other direction, we consider the two colored graphs $G$ and $H$ in Figure 3(b). Also let $T_1$ and $T_2$ be the two trees in Figure 3(c). Note that $T_1$ and $T_2$ are the subtrees of red vertices in $G$ and $H$ respectively.

We will first check that $\mathrm{RePHINE}^{mSpec}$ cannot tell the difference between them. For the ease of notations, we will write $r = f_v(\mathrm{red})$ and $b = f_v(\mathrm{blue})$. It suffices for us to check this in the following three cases on $f_v$.

1. $r > b$: Recall that the induced function $F_e$ for max RePHINE is given by $F_e((v_1, v_2)) = \max(F_v(v_1), F_v(v_2))$ for an edge $(v_1, v_2)$ in the graph. Since $r > b$ and every vertex in $G$ (resp. $H$) is either red or connected to a red vertex, we conclude that $F_e$ is constant with value $r$ on the edges of both $G$ and $H$.

   Since $G$ and $H$ are both trees, $\mathrm{RePHINE}^{mSpec}(G, f_v)^1$ and $\mathrm{RePHINE}^{mSpec}(H, f_v)^1$ are both the empty list. It suffices for us to examine

$$\mathrm{RePHINE}^{mSpec}(G, f_v)^0 = \{(0, d_G(v), \alpha_G(v), \gamma_G(v), \rho_G(v)\}_{v \in V(G)}$$

   and

$$\mathrm{RePHINE}^{mSpec}(H, f_v)^0 = \{(0, d_H(w), \alpha_H(w), \gamma_H(w), \rho_H(w)\}_{w \in V(H)}.$$

   Clearly, $G$ and $H$ are isomorphic as uncolored graphs, and it therefore follows that the data $\rho_G(v)$ and $\rho_H(w)$ are both the same. Since $F_e$ is constant, it also follows that $\gamma_G(v)$ and $\gamma_H(w)$ are the same. For both $G$ and $H$, they will have exactly 1 blue vertex that dies at $\infty$, 1 blue vertex that dies at time $r$, and 10 red vertices that dies at time $r$. Hence, we conclude that their respective max RePHINE Spectral diagrams will be the same.

2. $r = b$: In this case, we completely lose the added information that there are blue vertices in the graph. Let $R$ be the same graph as $G$ but with every vertex colored red, then we clearly have that $\mathrm{RePHINE}^{mSpec}(G, f_v) = \mathrm{RePHINE}^{mSpec}(R, f_v) = \mathrm{RePHINE}^{mSpec}(H, f_v)$.

3. $r < b$: At time $r$, we observe that $G_r^{f_e}$ is the disjoint union of the tree $T_1$ and two vertices. On the other hand, $H_r^{f_e}$ is the disjoint union of the tree $T_2$ and two vertices. For both $G$ and $H$ at time $r$, there will be 10 red vertices that die at this time. Their $\alpha$-values will be the same (being $r$ itself). Their $\gamma$-values will also be equal to $r$ as $\gamma$ takes the minimum of the values of $f_e$ on edges adjacent to each vertex. Their $\rho$-values will also be the same. This is because we can compute and find that the characteristic polynomials of $\Delta_0(T_1)$ and $\Delta_0(T_2)$ are both equal to

$$x^{11} - 20x^{10} + 166x^9 - 748x^8 + 2014x^7 - 3368x^6 + 3525x^5 - 2264x^4 + 843x^3 - 160x^2 + 11x.$$

At time $b$, $G_b^{f_e} = G$ and $H_b^{f_e} = H$. For both $G$ and $H$, there will be 2 blue vertices that die at this time. Their $\alpha$-values will be the same (being $b$). Their $\gamma$-values will be both equal to $b$, and their $\rho$-values will be the same since, again, $G$ and $H$ are isomorphic as uncolored graphs.

At the time $\infty$, the remaining 1 red vertex in $G$ and $H$ will both die, and it is clear that their $\alpha, \gamma$, and $\rho$-values will be the same.

Now we will show that RePHINE can tell the difference between $G$ and $H$. Intuitively, the difference between the RePHINE diagram and the max spectral RePHINE diagram is that the former can assign the colors "red-blue" and "red-red" to the edges of $G$ and $H$ while the latter can only assign the color "red", so there is an extra degree of freedom in the RePHINE diagram. Now, let us set $r > b$ and $f_e(\text{red-blue}) = 1$ and $f_e(\text{red-red}) = 2$. Now, we have that $G_1^{f_e}$ is the disjoint union of 10 vertices and a path graph of 3 vertices (linking the two blue vertices to the middle red vertex). On the other hand, we have that $H_1^{f_e}$ is the disjoint union of 9 vertices and two separate path graphs of two vertices (each path links a blue vertex to a red vertex). Hence, we see that 1 red vertex and 1 blue vertex died in $G$ at time 1, but two red vertices died in $H$ at time 1. Hence, they have different RePHINE diagrams. $\square$

**Proposition 11.** $\text{RePHINE}^{Spec}$ *is strictly more expressive than* $\text{RePHINE}^{mSpec}$.

*Proof.* From Definition 6, we know that $\text{RePHINE}^{mSpec}$ is by construction a special case of $\text{RePHINE}^{Spec}$. Hence, $\text{RePHINE}^{Spec}$ has at least the same expressive power as $\text{RePHINE}^{mSpec}$. In the proof of Proposition 10, we know that $\text{RePHINE}^{mSpec}$ cannot tell the difference between $G$ and $H$ in Figure 3(b), but RePHINE can. Since $\text{RePHINE}^{Spec}$ is a generalization of RePHINE, it can certainly tell the difference between $G$ and $H$ too. We thus conclude that $\text{RePHINE}^{Spec}$ is strictly more expressive than $\text{RePHINE}^{mSpec}$. $\square$

**Proposition 12.** $\text{RePHINE}$ *is strictly more expressive than* $\text{RePHINE}^m$.

*Proof.* From Definition 4, we know that $\text{RePHINE}^m$ is by construction a special case of RePHINE. Hence, RePHINE has at least the same expressive power as $\text{RePHINE}^m$. In the proof of Proposition 10, we know that $\text{RePHINE}^{mSpec}$ cannot tell the difference between $G$ and $H$ in Figure 3(b), but RePHINE can. Since $\text{RePHINE}^{mSpec}$ is a generalization of $\text{RePHINE}^m$, $\text{RePHINE}^m$ itself certainly cannot tell the difference between $G$ and $H$ either. We thus conclude that RePHINE is strictly more expressive than $\text{RePHINE}^m$. $\square$

**Proposition 13.** PH *is strictly more expressive than* EC.

*Proof.* In the proof of Proposition 19, we showed that $\text{PH}(G, f_v, f_e) = \text{PH}(H, f_v, f_e)$ implies $\text{EC}(G, f_v, f_e) = \text{EC}(H, f_v, f_e)$. Thus, PH has at least the same expressive power as EC. Now consider the graphs $G$ and $H$ in Figure 4. Since all vertices have the same color, $f_v$ has to be constant, and the edge coloring function $f_e$ is also necessarily constant. Let $a$ be the constant value of $f_v$ and $b$ be the constant value of $f_e$, then we clearly have that

$$\chi(G_a^{f_v}) = 0 = \chi(H_a^{f_v}) \text{ and } \chi(G_b^{f_e}) = 0 = \chi(H_b^{f_e}).$$

Hence, their respective EC diagrams have to be the same. On the other hand, the zeroth Betti numbers (ie. the number of connected components) of $G$ and $H$ are clearly different, so we can find two different PH diagrams of $G$ and $H$. $\square$

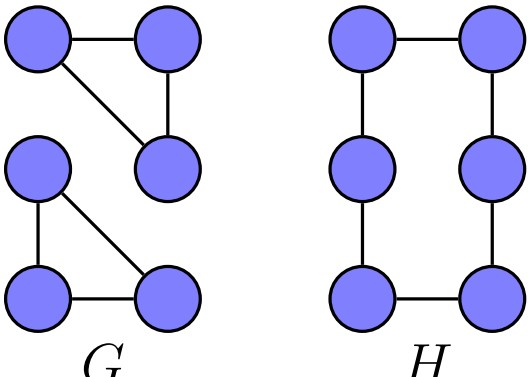

Figure 4: Illustration of graphs $G$ and $H$ that PH can tell apart but EC cannot.

**Proposition 14.** RePHINE$^m$ *is strictly more expressive than* EC$^m$.

*Proof.* By restricting to the first two components of the tuples provided by the max RePHINE diagram, we can use a similar proof in that of Proposition 19 to show that RePHINE$^m(G, f_v) =$ RePHINE$^m(H, f_v)$ implies EC$^m(G, f_v) = $ EC$^m(H, f_v)$. To show that RePHINE$^m$ is strictly more expressive than EC$^m$, we can use the same example as in Figure 4. The proof follows similarly to that of Proposition 13 because there is no difference between max RePHINE, RePHINE, and PH on graphs where all vertices have the same color. $\square$

A.2   PROOFS FOR SECTION 3.2

In this section, we will characterize the expressivity of EC on graphs. For the discussion of EC on simplicial complexes, please see Appendix C.

Before we state the proofs, we first remind the reader that in Section 3.2, we assumed that $G$ and $H$ have the same number of vertices (otherwise, they clearly are not isomorphic to each other).

*Proof of Proposition 1.* We will first show that the second and third items are equivalent. For each vertex color function $f : X \to \mathbb{R}$, we use $g_f : X \times X \to \mathbb{R}$ to denote its correspondent edge color function.

Suppose EC$^m(G, f)^1 = $ EC$^m(H, f)^1$ for all possible $f : X \to \mathbb{R}$. Recall by assumption $G$ and $H$ have the same number of vertices $n$. Let $f$ be an vertex color function such that $0 < f(a) < f(b) < \min_{c_i \in C - \{a,b\}} f(c_i)$, then we have that

$$n - \#E_G(a, a) = \chi(G_a^{g_f}) = \chi(H_a^{g_f}) = n - \#E_H(a, a)$$

Hence $\#E_G(a, a) = \#E_H(a, a)$. By choosing another appropriate vertex color function, we can similarly show that $\#E_G(b, b) = \#E_H(b, b)$. Now, if we look back on the function $f$, we have that

$$n - \#E_G(a, a) - \#E_G(a, b) - \#E_G(b, b) = \chi(G_b^{g_f})$$

$$= \chi(H_b^{g_f}) = n - \#E_H(a, a) - \#E_H(a, b) - \#E_H(b, b).$$

This implies that $\#E_G(a, b) = \#E_H(a, b)$ since we already know that $\#E_G(a, a) = \#E_H(a, a)$ and $\#E_G(b, b) = \#E_H(b, b)$.

Conversely, suppose $\#E_G(a, b) = \#E_H(a, b)$ for all colors $a, b \in X$. Let $f$ be an edge color filtration function such that $0 < f(d_1) \le ... \le f(d_s)$, where $d_i$ is a relabeling of the colors $c_1, ..., c_s$ by this ordering.

Since $G$ and $H$ have the same number of vertices $n$, we have that $\chi(G_0^{g_f}) = n = \chi(H_0^{g_f})$. Thus, it suffices for us to show that $\chi(G_t^{g_f}) = \chi(H_t^{g_f})$ for all $t > 0$. Indeed,

$$\chi(G_t^{g_f}) = n - \sum_{i \leq j \text{ s.t. } f(d_i), f(d_j) \leq t} \#E_G(d_i, d_j)$$

$$= n - \sum_{i \leq j \text{ s.t. } f(d_i), f(d_j) \leq t} \#E_H(d_i, d_j)$$

$$= \chi(H_t^{g_f}).$$

Hence, we have that the second and third items are equivalent. Finally, we observe that clearly (1) implies (2) because the edge function induced by the max EC diagram is a special case of a symmetric edge coloring function. An argument very similar to the proof of (3) implies (2) will also show that (3) implies (1). This is because for all $t > 0$,

$$\chi(G_t^g) = n - \sum_{i \leq j \text{ s.t. } g(d_i, d_j) \leq t} \#E_G(d_i, d_j)$$

$$= n - \sum_{i \leq j \text{ s.t. } g(d_i, d_j) \leq t} \#E_H(d_i, d_j)$$

$$= \chi(H_t^g).$$

$\square$

*Proof of Proposition 2.* There is no difference between the definition of the 0-th dimensional component of the EC diagram and the max EC diagram, so the first two items are equivalent.

For (1) implies (3), we can choose any injective vertex color function such that $f$ obtains its minimum at the color $c$, Then $G_{f(c)}^f$ and $H_{f(c)}^f$ will be $G(c)$ and $H(c)$, and hence their Euler characteristics will be the same. For any $a \neq b \in X$, we can choose an injective vertex color function $f$ such that $f$ is smallest at $a$ and second smallest at $b$. In this case, we have that

$$\chi(G_{f(b)}^f) = \chi(H_{f(b)}^f).$$

Here we have that

$$\chi(G_{f(b)}^f) = \chi(G(a)) - \#E_G(a, b) + \chi(G(b)),$$

$$\chi(H_{f(b)}^f) = \chi(H(a)) - \#E_H(a, b) + \chi(H(b)).$$

Hence, we conclude that $\#E_G(a, b) = \#E_H(a, b)$ for all $a \neq b \in X$. The proof of (3) implies (1) is similar to that in Proposition 1. $\square$

*Proof of Theorem 3.* The first two statements come directly from combining the statements of Proposition 1 and Proposition 2. The third statement follows from the observation that $\chi(G(c)) + \#E_G(c, c) = \#V_G(c)$. $\square$

A.3 PROOFS FOR SECTION 4

For ease of notation, we will omit the parameter $G$ in the RePHINE diagram in this section since we will only be discussing functions on the same graph. We will first verify that our definition of a metric $d_B^R$ in Definition 7 is actually a metric. To do this, we first note that the definition of $d_B^R$ can be elaborated as the following definition.

**Definition 11.** *Let* RePHINE$(G, f)$ *and* RePHINE$(G, g)$ *be the two associated RePHINE diagrams for $G$ respectively. We define the **bottleneck distance** as* $d_B^R(\text{RePHINE}(G, f), \text{RePHINE}(G, g)) := d_B^{R,0}(\text{RePHINE}(G, f)^0, \text{RePHINE}(G, g)^0) + d_B(\text{RePHINE}(G, f)^1, \text{RePHINE}(G, g)^1).$

*For the 0-th dimensional component, $d_B^{R,0}$ is defined as,*

$$d_B^{R,0}(\text{RePHINE}(G, f)^0, \text{RePHINE}(G, g)^0) := \inf_{\pi \in bijections} \max_{p \in \text{RePHINE}(G,f)^0} d(p, \pi(p)),$$

*where $d$ is defined as $d((b_0, d_0, \alpha_0, \gamma_0), (b_1, d_1, \alpha_1, \gamma_1)) = \max\{|b_1 - b_0|, |d_1 - d_0|\} + |\alpha_1 - \alpha_0| + |\gamma_1 - \gamma_0|$, and $\pi$ ranges over all bijections $\mathrm{RePHINE}(G, f)^0 \to \mathrm{RePHINE}(G, g)^0$. For the $1$-st dimensional component, $d_B$ is the usual bottleneck distance on $1$-dimensional persistence pairs.*

**Proposition 15.** $d_B^R$ *is a metric.*

*Proof.* Since the usual bottleneck distance is a metric, it suffices for us to check that $d_B^{R,0}$ is a metric.

**1. Non-degeneracy:** Clearly the function is non-negative, and choosing $\pi$ to be the identity bijection shows that $d_B^{R,0}(\mathrm{RePHINE}(f)^0, \mathrm{RePHINE}(f)^0) = 0$. For any pairs $\mathrm{RePHINE}(f)^0 \neq \mathrm{RePHINE}(g)^0$, the term $\max_{p \in \mathrm{RePHINE}(f)^0} d(p, \pi(p))$ will be greater than 0 for any choice of bijection $\pi$. Since there are only finitely many possible bijections $\pi$, the infimum $d_B^{R,0}(\mathrm{RePHINE}(f)^0, \mathrm{RePHINE}(g)^0)$ will be greater than 0.

**2. Symmetry:** Any bijection $\pi : \mathrm{RePHINE}(f)^0 \to \mathrm{RePHINE}(g)^0$ corresponds exactly to a bijection $\pi^{-1} : \mathrm{RePHINE}(g)^0 \to \mathrm{RePHINE}(f)^0$. Hence, the definition of $d_B^{R,0}$ is symmetric.

**3. Triangle Inequality:** Let $\mathrm{RePHINE}(f)^0, \mathrm{RePHINE}(g)^0, \mathrm{RePHINE}(h)^0$ be the vertex components of RePHINE diagrams on $G$. Suppose $\sigma_1 : \mathrm{RePHINE}(f)^0 \to \mathrm{RePHINE}(h)^0$ is a bijection that achieves the infimum labeled in the definition of $d_B^{R,0}$. In other words,

$$d_B^{R,0}(\mathrm{RePHINE}(f)^0, \mathrm{RePHINE}(h)^0) = \max_{p \in \mathrm{RePHINE}(f)^0} d(p, \sigma_1(p)).$$

Suppose $\tau_1, \tau_2$ are bijections from $\mathrm{RePHINE}(f)^0 \to \mathrm{RePHINE}(g)^0$ and $\mathrm{RePHINE}(g)^0 \to \mathrm{RePHINE}(h)^0$ respectively. We then have that,

$$\max_{p \in \mathrm{RePHINE}(f)^0} d(p, \sigma_1(p)) \leq \max_{p \in \mathrm{RePHINE}(f)^0} d(p, \tau_1(p)) + d(\tau_1(p), \tau_2(\tau_1(p)))$$

$$\text{Triangle Inequality for } d$$

$$\leq \max_{p \in \mathrm{RePHINE}(f)^0} d(p, \tau_1(p)) + \max_{q \in \mathrm{RePHINE}(g)^0} d(q, \tau_2(q)).$$

For the sake of paragraph space, we write $A = d_B^{R,0}(\mathrm{RePHINE}(f)^0, \mathrm{RePHINE}(h)^0)$. Taking infimum over all possible $\tau_1$ and over $\tau_2$ gives us that

$$A = d_B^{R,0}(\mathrm{RePHINE}(f)^0, \mathrm{RePHINE}(h)^0)$$
$$\leq \inf_{\tau_1, \tau_2} \max_{p \in \mathrm{RePHINE}(f)^0} d(p, \tau_1(p)) + \max_{q \in \mathrm{RePHINE}(g)^0} d(q, \tau_2(q))$$
$$\leq \inf_{\tau_1 \in \text{bijection}} \max_{p \in \mathrm{RePHINE}(f)^0} d(p, \tau_1(p)) + \inf_{\tau_2 \in \text{bijection}} \max_{q \in \mathrm{RePHINE}(g)^0} d(q, \tau_2(q))$$
$$\leq d_B^{R,0}(\mathrm{RePHINE}(f)^0, \mathrm{RePHINE}(g)^0) + d_B^{R,0}(\mathrm{RePHINE}(g)^0, \mathrm{RePHINE}(h)^0).$$

This shows that $d_B^{R,0}$ satisfies the triangle inequality. $\qquad\square$

To prove Theorem 4, we first define a technical construction as follows.

**Definition 12.** *Given a graph $G$ and functions $(f_v, f_e)$ on $X$. This induces functions $(F_v, F_e)$ as defined in Definition 1. From here, we construct a pairing between $\mathrm{RePHINE}(f)^0$ and $(v, e) \in V(G) \times \{0\} \cup E(G)$ as follows.*

1. *For every almost hole $(0, d)$ that occurs in the edge filtration by $F_e$, this corresponds to the merging of two connected components represented by vertices $v_i$ and $v_j$.*

2. *We assign to $(0, d)$ the vertex that has greater value under $\alpha$. If there is a tie, we assign the vertex that has the lower value under $\gamma$. If there is a further tie, we will be flexible in how we assign them in the proof of the stability of the RePHINE diagram.*

3. *The occurrence of an almost hole $(0, d)$ is caused by an edge $e$ whose value under $f_e$ is $d$ that merges two connected components. We assign this edge to $(0, d)$. If there are multiple such edges, we will be flexible in how we assign them in how we assign them in the proof of the stability of the RePHINE diagram.*

4. *For the real holes, we assign them with the vertices left. The edge takes an uninformative value (ie. $0$).*

*Note that for any vertex $v$ that dies at finite time $d$, its associated edge $e = (v_1, v_2)$ satisfies $f_e(c(v_1), c(v_2)) = d(v)$.*

We will now state and prove two propositions that will directly imply Theorem 4. Our proof adapts the methods presented in Skraba & Turner (2021).

**Proposition 16.** *Suppose $f_e = g_e = h$ for some edge coloring function $h$, then*

$$d_B^{R,0}(\text{RePHINE}(f_v, h)^0, \text{RePHINE}(g_v, h)^0) \leq ||f_v - g_v||_\infty.$$

*Proof.* Let $h_v^t(x) = (1-t)f_v(x) + tg_v(x)$. Also let $H_v^t : G \to \mathbb{R}$ be the induced function of $h_v^t$ on $G$, in the sense of Definition 1. We can divide $[0, 1]$ into finite intervals $[t_0, t_1], [t_1, t_2], ..., [t_n, t_{n+1}]$, where $t_0 = 0, t_{n+1} = 1, t_0 < t_1 < ... < t_{n+1}$, such that for all $t \in [t_i, t_{i+1}]$ and all simplicies $x, y \in G$, either

$$H_v^t(x) - H_v^t(y) \leq 0 \text{ or } \geq 0 \quad (\dagger).$$

To be clear on the wording, this means that we cannot find $s, s' \in [t_i, t_{i+1}]$ such that $H_v^s(x) > H_v^s(y)$ but $H_v^{s'}(x) < H_v^{s'}(y)$.

For all $s_1, s_2 \in [t_i, t_{i+1}]$, we claim that we can use Definition 12 to produce the same list of pairs $(v, e)$ (with some flexible adjustments at endpoints if needed).

Since $f_e = g_e$, the list of death times and order of edges that appear do not change, what could change is which vertex to kill off at the time stamp. Let us now order the finite death times (ie. those corresponding to almost holes), accounting for multiplicity, as $d_1 \leq d_2 \leq ... \leq d_n < \infty$. Now we observe that

1. At $d_1$, $\text{RePHINE}(h_v^{s_1}, h)$ and $\text{RePHINE}(h_v^{s_2}, h)$ will be merging the same two connected components with vertex representatives $v$ and $w$. For the RePHINE diagram at $s_1$ (resp. $s_2$), we choose which vertex to kill off based on which vertex has a higher value under $H_v^{s_1}$ (resp. $H_v^{s_2}$). By $(\dagger)$, we will be killing off the same vertex. If there happens to be a tie of $\alpha$ values, we will still kill off the same vertex in the comparison of $\gamma$ values since $f_e = g_e$. Finally, if there is a tie of $\gamma$ values, we make the flexible choice to kill off the same vertex.

   Since $f_e = g_e$, the edge associated to this vertex can be chosen to be the same. Hence, $\text{RePHINE}(h_v^{s_1}, h)$ and $\text{RePHINE}(h_v^{s_2}, h)$ will produce the same pair $(v, e)$ at time $d_1$.

2. Suppose that up to the $i$-th death, both RePHINE diagrams are producing the same pairs and merging the same components. For the $i + 1$-death, the RePHINE diagrams at both $s_1$ and $s_2$ will be merging the same two components $v'$ and $w'$. The same argument as the case for $d_1$ shows that they will produce the same pair of vertex and edge.

3. After we go through all finite death times, both RePHINE diagrams will have the same list of vertices that are not killed off, which are then matched to real holes.

This proves the claim above. From triangle inequality, we know that $d_B^{R,0}(\text{RePHINE}(f_v, h)^0, \text{RePHINE}(g_v, h)^0)$ is bounded by the term

$$\sum_{i=0}^{n} d_B^{R,0}(\text{RePHINE}(h_v^{t_i}, h)^0, \text{RePHINE}(h_v^{t_{i+1}}, h)^0).$$

For each summand on the right, we assign a bijection from $\text{RePHINE}(h_v^{t_i}, h)^0$ to $\text{RePHINE}(h_v^{t_{i+1}}, h)^0$ as follows - using our previous claim, we send $(0, d, \alpha, \gamma) \in \text{RePHINE}(h_v^{t_i}, h)^0$ to the pair in $\text{RePHINE}(h_v^{t_{i+1}}, h)^0$ that correspond to the same $(v, e)$. For the sake of paragraph space, we write $A_i = d_B^{R,0}(\text{RePHINE}(h_v^{t_i}, h)^0, \text{RePHINE}(h_v^{t_{i+1}}, h)^0)$ and

see that

$$
\begin{aligned}
A_i &= d_B^{R,0}(\text{RePHINE}(h_v^{t_i}, h)^0, \text{RePHINE}(h_v^{t_{i+1}}, h)^0) \\
&\leq \max_{(v,e)} |d_{t_{i+1}}(v) - d_{t_i}(v)| + |\alpha_{t_{i+1}}(v) - \alpha_{t_i}(v)| + |\gamma_{t_{i+1}}(v) - \gamma_{t_i}(v)| \\
&= \max_{(v,e)} 0 + |\alpha_{t_{i+1}}(v) - \alpha_{t_i}(v)| + 0 \qquad\qquad \text{Since } f_e = g_e \\
&= \max_v |\alpha_{t_{i+1}}(v) - \alpha_{t_i}(v)| \\
&= \max_w |h_v^{t_{i+1}}(c(w)) - h_v^{t_i}(c(w))| \\
&= \max_w |(1 - t_{i+1})f_v(c(w)) + t_{i+1}g_v(c(w)) - (1 - t_i)f_v(c(w)) - t_i g_v(c(w))| \\
&= \max_w |(t_i - t_{i+1})f_v(c(w)) + (t_{i+1} - t_i)g_v(c(w))| \\
&= \max_w (t_{i+1} - t_i)|f_v(c(w)) - g_v(c(w))| \\
&\leq (t_{i+1} - t_i)\|f_v - g_v\|_\infty.
\end{aligned}
$$

Hence, we have that

$$
d_B^{R,0}(\text{RePHINE}(f_v, h)^0, \text{RePHINE}(g_v, h)^0) \leq \sum_{i=0}^n A_i
$$

$$
\leq \sum_{i=0}^n (t_{i+1} - t_i)\|f_v - g_v\|_\infty
$$

$$
= \|f_v - g_v\|_\infty.
$$

$\square$

**Remark 1.** *In the proof of the previous proposition, we claimed that we can assign the same vertex-edge pair to each death time for both filtration at $t_i$ and $t_{i+1}$. This may seem contradictory at first, as this seems to suggest that, by connecting the endpoints of the interval, RePHINE would assign the same vertices on $G$ as real holes regardless of the choice of functions. However, in our proof, the choice of vertex-edge assignment on $t_i \in [t_i, t_{i+1}]$ need not be the same as the choice of vertex-edge assignment on $t_i \in [t_{i-1}, t_i]$. This is what we meant by "flexibility" in Definition 12, the key point is that both choices will give the same RePHINE diagram.*

**Proposition 17.** *Suppose $f_v = g_v = h$ for some vertex coloring function $h$, then*

$$
d_B^{R,0}(\text{RePHINE}(h, f_e)^0, \text{RePHINE}(h, g_e)^0) \leq 2\|f_e - g_e\|_\infty.
$$

*Proof.* Let $h_e^t(x) = (1 - t)f_e(x) + t g_e(x) : X \times X \to \mathbb{R}_{>0}$ be as in the previous proof. Also let $H_e^t : G \to \mathbb{R}_{\geq 0}$ denote the induced function on $G$ in the sense of Definition 1. We can divide $[0, 1]$ into finite intervals $[t_0, t_1], [t_1, t_2], ..., [t_n, t_{n+1}]$, where $t_0 = 0, t_{n+1} = 1, t_0 < t_1 < ... < t_{n+1}$, such that for all $t \in [t_i, t_{i+1}]$ and all edges $x, y \in G$, either

$$
H_e^t(x) - H_e^t(y) \leq 0 \text{ or } \geq 0 \quad (\dagger).
$$

For all $s_1, s_2 \in [t_i, t_{i+1}]$, we claim that we can use Definition 12 to produce the same list of pairs $(v, e)$ (with some flexible adjustments at endpoints if needed).

The death times for the RePHINE diagrams at $s_1$ and $s_2$ may be different. Let us write $d_1^{s_1} \leq ... \leq d_n^{s_1}$ (with multiplicity) to indicate all the finite death times for $s_1$, and similarly we write $d_1^{s_2} \leq ... \leq d_n^{s_2}$ for $s_2$ (with reordering allowed for deaths that occur at the same time). We claim that the corresponding $(v, e)$ produced at $d_1^{s_2}$ and $d_i^{s_2}$ can be chosen to be the same.

1. For each death time that occurs, we are free to choose any of the merging of two components that occurred at that time to be assigned to that death time.

2. At $d_1^{s_1}$, the death occurs between the merging of two vertices $v$ and $w$ by an edge $e$ such that $H_e^{s_1}(e) = d_1^{s_1}$. If $H_e^{s_2}(e) = d_1^{s_2}$, then we can choose the first death to occur with the same edge $e$ between $v$ and $w$.

Otherwise, suppose $H_e^{s_2}(e) > d_1^{s_2}$. There exists an edge $e'$ such that $H_e^{s_2}(e') = d_1^{s_2}$, so $H_e^{s_2}(e) > H_e^{s_2}(e')$. By (†), this means that $H_e^{s_1}(e) \geq H_e^{s_1}(e')$, which implies that $H_e^{s_1}(e') = d_1^{s_1}$. We instead choose the first death in $d_1^{s_1}$ to occur with the edge $e'$ between its adjacent vertices.

In either case, we see that at the first death time, we can choose an assignment such that the RePHINE diagrams at $s_1$ and $s_2$ are merging the same two connected components $a, b$ by the same edge. Now we will show that they will kill off the same vertex. Since $f_v = g_v$, the first comparison will always give the same result. If there is a tie, then we are comparing $f_e$ and $g_e$. Suppose for contradiction, that without loss, $a$ has lower $\gamma$ value at $s_1$ and $b$ has lower $\gamma$ value at $s_2$. This means that there exists an edge $e_a$ adjacent to $a$ such that

$$H_e^{s_1}(e_a) < H_e^{s_1}(e), \text{ for all } e \text{ adjacent to } b.$$

Now, since $b$ has lower $\gamma$ value at $s_2$, this means that there exists an edge $e_b$ adjacent to $b$ such that

$$H_e^{s_2}(e_b) < H_e^{s_2}(e), \text{ for all } e \text{ adjacent to } a.$$

In particular, this means that $H_e^{s_2}(e_b) < H_e^{s_2}(e_a)$ and $H_e^{s_1}(e_b) > H_e^{s_1}(e_a)$, which violates (†). Hence, we will have a consistent vertex to kill off. Finally, if there is a tie, then we flexibly choose the same vertex to kill off. Since we are comparing the same edge, there is a canonical edge associated too.

Thus, we have shown that $d_1^{s_1}$ and $d_2^{s_2}$ can be chosen to give the same vertex edge pair.

3. Inductively, suppose that up to the $i$-th death, both RePHINE diagrams are producing the same pairs and merging the same components.

   For the $i + 1$-th death, $d_{i+1}^{s_1}$ occurs between the merging of two connected components $C_1$ and $C_2$ by an edge $e$ such that $H_e^{s_1}(e) = d_{i+1}^{s_1}$. Now if $H_e^{s_2}(e) = d_{i+1}^{s_2}$, then by our inductive hypothesis we can choose both filtration so that they would be merging the same connected components.

   Now suppose $H_e^{s_2}(e) \neq d_{i+1}^{s_2}$. By the inductive hypothesis, it cannot be lower, so $H_e^{s_2}(e) > d_{i+1}^{s_2}$. In this case, we look at $d_{i+1}^{s_2}$ itself, which also occurs with an edge $e'$ that merges connected components $C_1'$ and $C_2'$. Hence, we have that

   $$H_e^{s_2}(e) > H_e^{s_2}(e') = d_{i+1}^{s_2}.$$

   By (†), this means that

   $$H_e^{s_1}(e) \geq H_e^{s_2}(e').$$

   By our inductive hypothesis, this edge $e'$ cannot have occurred in prior deaths, hence we have that $H_e^{s_2}(e') = d_{i+1}^{s_1}$, and the same arguments as the base case follow through.

   In either case, we see that at the $i + 1$-th death time, we can choose an assignment such that the RePHINE diagrams at $s_1$ and $s_2$ are merging the same two connected components $a, b$ by the same edge. Similar to our discussion in the base case, the vertex-edge pair produced would be consistent.

4. After we go through all finite death times, both RePHINE diagrams will have the same list of vertices that are not killed off, which are then matched to real holes.

This proves the claim above. Now, from triangle inequality, we again have that $d_B^{R,0}(\text{RePHINE}(h, f_e)^0, \text{RePHINE}(h, g_e)^0)$ is bounded by the sum

$$\sum_{i=0}^{n} d_B^{R,0}(\text{RePHINE}(h, h_e^{t_i})^0, \text{RePHINE}(h, h_e^{t_{i+1}})^0).$$

For each summand on the right, we assign a bijection with the exact same strategy as the proof of the previous proposition. For the sake of paragraph space, we write $A_i =$

$d_B^{R,0}(\text{RePHINE}(h, h_e^{t_i})^0, \text{RePHINE}(h, h_e^{t_{i+1}})^0)$ and compute that

$$\begin{aligned}
A_i &= d_B^{R,0}(\text{RePHINE}(h, h_e^{t_i})^0, \text{RePHINE}(h, h_e^{t_{i+1}})^0) \\
&\leq \max_{(v,e)} |d_{t_{i+1}}(v) - d_{t_i}(v)| + |\alpha_{t_{i+1}}(v) - \alpha_{t_i}(v)| + |\gamma_{t_{i+1}}(v) - \gamma_{t_i}(v)| \\
&= \max_{(v,e)} |d_{t_{i+1}}(v) - d_{t_i}(v)| + 0 + |\gamma_{t_{i+1}}(v) - \gamma_{t_i}(v)| && \text{Since } f_v = g_v \\
&= \max_{(v,e)} |H_e^{t_{i+1}}(e) - H_e^{t_i}(e)| + |\gamma_{t_{i+1}}(v) - \gamma_{t_i}(v)| \\
&\leq (t_{i+1} - t_i)||f_e - g_e||_\infty + \max_v |\gamma_{t_{i+1}}(v) - \gamma_{t_i}(v)|.
\end{aligned}$$

We claim that $|\gamma_{t_{i+1}}(v) - \gamma_{t_i}(v)| \leq ||h_e^{t_{i+1}} - h_e^{t_i}||_\infty$. Indeed, without loss let us say $\gamma_{t_{i+1}}(v) \geq \gamma_{t_i}(v)$. Let $e_i$ be the edge adjacent to $v$ that has minimum value under $H_e^{t_i}$, then this means that

$$\begin{aligned}
|\gamma_{t_{i+1}}(v) - \gamma_{t_i}(v)| &= \gamma_{t_{i+1}}(v) - \gamma_{t_i}(v) \\
&= \gamma_{t_{i+1}}(v) - H_e^{t_i}(e_i) \\
&\leq H_e^{t_{i+1}}(e_i) - H_e^{t_i}(e_i) && \text{Since } \gamma_{t_{i+1}}(v) \text{ is minimum} \\
&\leq ||H_e^{t_{i+1}} - H_e^{t_i}||_\infty \\
&\leq ||h_e^{t_{i+1}} - h_e^{t_i}||_\infty \\
&\leq (t_{i+1} - t_i)||f_e - g_e||_\infty.
\end{aligned}$$

Thus, we have that

$$A_i = d_B^{R,0}(\text{RePHINE}(h, f_e)^0, \text{RePHINE}(h, g_e)^0) \leq 2(t_{i+1} - t_i)||f_e - g_e||_\infty.$$

Hence, we have that

$$\begin{aligned}
d_B^{R,0}(\text{RePHINE}(h, f_e)^0, \text{RePHINE}(h, g_e)^0) &\leq \sum_{i=0}^n A_i \\
&\leq \sum_{i=0}^n 2(t_{i+1} - t_i)||f_e - g_e||_\infty \\
&= 2||f_e - g_e||_\infty.
\end{aligned}$$

$\square$

Now we finally give a proof of Theorem 4.

*Proof of Theorem 4.* On the 1-dimensional components of the RePHINE diagram, we have the usual bottleneck distance. Cohen-Steiner et al. (2006) gives a standard bound on this term by $d_B(\text{RePHINE}(f)^1, \text{RePHINE}(g)^1) \leq ||f_e - g_e||_\infty$. The theorem then follows from the triangle inequality, the inequality in the previous sentence, and the previous two propositions. $\square$

From Theorem 4, we also prove the corollary on max RePHINE diagrams.

*Proof of Corollary 1.* It suffices for us to check that induced functions $f_e$ and $g_e$ satisfy $||f_e - g_e||_\infty \leq ||f_v - g_v||_\infty$. For any $(c_1, c_2) \in X \times X$, we have that

$$|f_e(c_1, c_2) - g_e(c_1, c_2)| = |\max(f_v(c_1), f_v(c_2)) - \max(g_v(c_1), g_v(c_2))|$$

If $c_1$ maximizes both terms or $c_2$ maximizes both terms, then we are done. Otherwise, without loss let us suppose that $\max(f_v(c_1), f_v(c_2)) = f_v(c_1)$ and $\max(g_v(c_1), g_v(c_2)) = g_v(c_2)$. Then in particular we have that

$$\begin{aligned}
f_v(c_1) - g_v(c_2) &\leq f_v(c_1) - g_v(c_1), \\
g_v(c_2) - f_v(c_1) &\leq g_v(c_2) - f_v(c_2).
\end{aligned}$$

Thus, we have that

$$|f_e(c_1, c_2) - g_e(c_1, c_2)| = |\max(f_v(c_1), f_v(c_2)) - \max(g_v(c_1), g_v(c_2))| \leq ||f_v - g_v||_\infty.$$

$\square$

Now we show that $d_B^{\mathrm{Spec}\,R}$ is a metric and that $\mathrm{RePHINE}^{Spec}$ is locally stable under this metric.

**Proposition 18.** $d_B^{\mathrm{Spec}\,R}$ *is a metric.*

*Proof.* Since we showed $d$ from Definition 7 is a metric, it suffices for us to check in this proposition that $d^{\mathrm{Spec}}$ is a metric. To reiterate the definition of $d^{\mathrm{Spec}}$, given a list $L$ of non-zero eigenvalues with length $n$, we define an embedding $\phi(L) \in \ell^1(\mathbb{N})$ where $\phi(L)$ where the first $n$-elements in the sequence are $L$ sorted in ascending order and the rest are zeroes. This embedding is clearly injective on the lists of non-zero eigenvalues. For two lists $\rho_0, \rho_1$, we define

$$d^{Spec}(\rho_0, \rho_1) = ||\phi(\rho_0) - \phi(\rho_1)||_1.$$

The fact that $d^{Spec}$ is a metric now follows from the fact that $\ell^1(\mathbb{N})$ is a metric space under its $\ell^1$-distance and $\phi$ is injective. $\square$

*Proof of Theorem 5.* We will again split this into two cases where $f_e = g_e$ and $f_v = g_v$ respectively.

Suppose again that $f_e = g_e = h$, let us try to follow the proof of Proposition 16 to give an idea on why using this method falls apart. let $h_v^t(x) = (1 - t)f_v(x) + tg_v(x)$ and $H_v^t : G \to \mathbb{R}$ be the induced function of $h_v^t$ on $G$, in the sense of Definition 1. We can again divide $[0, 1]$ into finite intervals $[t_0, t_1], [t_1, t_2], ..., [t_n, t_{n+1}]$, where $t_0 = 0, t_{n+1} = 1, t_0 < t_1 < ... < t_{n+1}$, such that for all $t \in [t_i, t_{i+1}]$ and all simplicies $x, y \in G$, either

$$H_v^t(x) - H_v^t(y) \le 0 \text{ or } \ge 0 \quad (\dagger).$$

For all $s_1, s_2 \in [t_0, t_1]$, we can again use Definition 12 to produce the same list of pairs $(v, e)$ (vertex to edge identification). Previously, by choosing $s_1 = t_0 = 0$ and $s_2 = t_1$, we were able to obtain a reasonable bound on the bottleneck distance for $\mathrm{RePHINE}$ in terms of the $L^\infty$ norms of $h_v^{t_0}$ and $h_v^{t_1}$. We could do this for $\mathrm{RePHINE}$ because the $b, d, \alpha, \gamma$ parameters of $\mathrm{RePHINE}$ are all not sensitive to the loss of injectivity. However, the $\rho$-parameter in $\mathrm{RePHINE}^{Spec}$ is sensitive to the loss of injectivity, as seen in Example 1. Moreover, the way we constructed the division of $[0, 1]$ indicates that we are forced to cross some time stamps $t$ in $[0, 1]$ where $h_v^t$ is no-longer injective.

However, we observe that clearly we could get the desired bound

$$d_B^{\mathrm{Spec}\,R,0}(\mathrm{RePHINE}^{Spec}(f_v, h)^0, \mathrm{RePHINE}^{Spec}(g_v, h)^0) \le ||f_v - g_v||_\infty,$$

provided that the following more restrictive condition holds - $h_v^t$ is injective for all $t \in [0, 1]$. The bounds on the $b, d, \alpha, \gamma$ parameters evidently follows from the same proof of Proposition 16. For the bound of $\rho$, we observe that in the production of the vertex-edge pairs $(v, e)$ in the proof of Proposition 16, we can choose the order of vertex deaths to be the same for both $(f_v, h) = (h_v^0, h)$ and $(g_v, h) = (h_v^1, h)$. Furthermore, the condition that $h_v^t$ is injective for all $t \in [0, 1]$ means that the ordering of colors in $X$ given by $f_v$ and $g_v$ respectively are exactly the same. Furthermore, both orderings are strict as they are injective. Thus, the component that the vertices die in at each time are also the same. What this effectively means is that, $\rho_f(v) = \rho_g(v)$ for all $v \in V$ (after choosing the $(v, e)$ identification). Thus, we would obtain the same bound.

Suppose $f_v = g_v = h$, then we note that an analogous argument would work to show the bound

$$d_B^{\mathrm{Spec}\,R,0}(\mathrm{RePHINE}^{Spec}(h, f_e)^0, \mathrm{RePHINE}^{Spec}(h, g_e)^0) \le 2||f_e - g_e||_\infty,$$

if we impose the condition that $h_e^t$ is injective for all $t \in [0, 1]$ in the context of the proof for Proposition 17.

We still need to check what happens for $d_B^{\mathrm{Spec}\,R,1}$, which is no longer the usual bottleneck distance. If $f_e = g_e = h$ and $h_v^t$ is injective for all $t \in [0, 1]$, then the 1st dimensional component of $\mathrm{RePHINE}^{Spec}(f_v, h)$ and $\mathrm{RePHINE}^{Spec}(g_v, h)$ would quite literally be identical. If $f_v = g_v = h$, and $h_e^t$ is injective for all $t \in [0, 1]$, then a similar argument as in Proposition 17 would show that

$$d_B^{\mathrm{Spec}\,R,1}(\mathrm{RePHINE}^{Spec}(h, f_e)^1, \mathrm{RePHINE}^{Spec}(h, g_e))^1 \le ||f_e - g_e||_\infty.$$

The idea is that the only obstruction to this bound was the presence of the $\rho$-parameter, which we could always choose the presence of cycles to have the same strict order with the same graph components showing up.

Thus, we have proven the following result - let $f = (f_v, f_e)$ and $g = (g_v, g_e)$, suppose $h_v^t(x) = (1-t)f_v(x) + tg_v(x)$ and $h_e^t(x) = (1-t)f_e(x) + tg_e(x)$ are injective for all $t \in [0,1]$, then

$$d_B^{\mathrm{Spec}\,R}(\mathrm{RePHINE}^{Spec}(f), \mathrm{RePHINE}^{Spec}(g)) \le 3||f_e - g_e||_\infty + ||f_v - g_v||_\infty.$$

It remains for us to show that the conditions on $h_v^t$ and $h_e^t$ are locally satisfied. However, we note that clearly $\mathrm{Conf}_{n_v}(\mathbb{R})$ and $\mathrm{Conf}_{n_e}(\mathbb{R}_{>0})$ are both locally convex, which is the same as imposing the hypothesis to obtain this bound. Thus, we have proven that $\mathrm{RePHINE}^{Spec}$ is locally stable. $\quad\square$

We remark that $\mathrm{Conf}_n(X)$ is an open subset of $X^n$, so $f = (f_v, f_e)$ would still be locally stable in $(\mathbb{R})^{n_v} \times (\mathbb{R}_{>0})^{n_e}$ provided $f_v$ and $f_e$ are both injective. Now we show that the injectivity assumption is necessary for local stability and $\mathrm{RePHINE}^{Spec}$ is not globally stable in general.

**Example 1.** *Let $G$ be the path graph on $4$ vertices colored in the order red, blue, blue, red. Let $f_v = g_v$ be any functions. Let $g_e$ be constant with value $1$, and $f_e$ be given by $f_e(red, blue) = 1$ and $f_e(blue, blue) = 1 - \epsilon$ for $\epsilon > 0$ ($f_e(red, red)$ can be any value). $\mathrm{RePHINE}^{Spec}(G, f)^0$ has $1$ tuple representing $1$ blue vertex that dies at time $t = 1 - \epsilon$ whose $\rho$-parameter is $\{2\}$. In contrast, every tuple of $\mathrm{RePHINE}^{Spec}(G, g)^0$ has its $\rho$-parameter equal to the list $L = \{2, 2 \pm \sqrt{2}\}$. No matter how small $\epsilon > 0$ is, the distance between their $\mathrm{RePHINE}^{Spec}$ diagrams is bounded below by*

$$d_B^{\mathrm{Spec}\,R}(\mathrm{RePHINE}^{Spec}(G, f), \mathrm{RePHINE}^{Spec}(G, g)) \ge d^{Spec}(\{2\}, L) > 0.$$

*We also note that $\mathrm{RePHINE}^{Spec}$ is not globally stable, even with the injectivity assumption. A counter-example can be found by changing $g_e$ in Example 1 to the function given by $g_e(red, red) = f_e(red, red)$, $g_e(red, blue) = 1$, and $g_e(blue, blue) = 1 + \epsilon$.*

### A.4 PROOFS FOR SECTION 5

*Proof of Proposition 3.* To check that the graphs are the same, we wish to check that they have the same vertex set and edge sets. Indeed, let $t \in \mathbb{R}$, then

$$V(G_t^{f_G} \square H_t^{f_H}) = \{(g, h) \mid f_G(c(g)) \le t \text{ and } f_H(c(h)) \le t\}$$

$$= \{(g, h) \mid \max(f_G(c(g)), f_H(c(h))) \le t\} = \{(g, h) \mid F((g, h)) \le t\} = V((G\square H)_t).$$

Now to check edges, we have that for $(g_1, h_1), (g_2, h_2) \in V(G_t \square H_t) = V((G\square H)_t)$. Now suppose there is an edge between $(g_1, h_1)$ and $(g_2, h_2)$ in $(G\square H)_t^f \subseteq G\square H$. Since this is an edge in $G\square H$ it means either of two following cases

1. $g_1 = g_2$ and $h_1 \sim h_2$ in $H$. Now if we can show that $h_1 \sim h_2$ in $H_t$, then this will give an edge between $(g_1, h_1)$ and $(g_2, h_2)$ in $G_t^{f_G} \square H_t^{f_H}$. Indeed, that just comes from the definition of a vertex filtration as the colors on $h_1, h_2$ are both less than or equal to $t$ under $f_H$.

2. $h_1 = h_2$ and $g_1 \sim g_2$ in $G$. The argument is nearly identical to the first case.

Conversely, suppose there is an edge between $(g_1, h_1)$ and $(g_2, h_2)$ in $G_t^{f_G} \square H_t^{f_H}$. Then, since $F((g_1, h_1)) \le t$ and $F((g_2, h_2)) \le t$ (as they have the same vertex set), it follows that this edge has to be in $(G\square H)_t$ by definition of vertex filtration. $\quad\square$

*Proof of Proposition 4.* To check that the graphs are the same, we wish to check that they have the same vertex set and edge sets for all $t \ge 0$. The vertex set at time $t$ for $t \ge 0$ is always the collection of all vertices. The proof for the edge sets follows similarly to the proof in Proposition 3. $\quad\square$

*Proof of Proposition 5.* Claim (1) is obvious. It suffices for us to verify Claim (2) since Claim (3) follows by a symmetric argument. For the ease of notations, let us write $n = h_t^b$ and $m = g_t^d$. Since the number of vertices still alive in $H_t$ corresponds to its number of connected components, we can

write $C_1, ..., C_n$ as the connected components of $H_t$. Now we have that, for all $s \in (t - \epsilon, t + \epsilon)$ for $\epsilon > 0$ sufficiently small,

$$(G \square H)_s = G_s \square H_t = G_s \square (\bigsqcup_{i=1}^{n} C_i) = \bigsqcup_{i=1}^{n} G_s \square C_i.$$

The number of vertices that dies at time $t$ is additive under this disjoint union, so it suffices for us to show that the number of vertices that die in $G_t \square C_i$ at time $t$ is $m = g_t^d$. Now indeed, write $D_1, ..., D_\ell$ as the connected components of $G_{t-\epsilon}$. However, at time $t$, the connected components $D_j$ and $D_k$ are merged if and only if the connected components $D_j \square C_i$ and $D_k \square C_i$ are merged. This proves the desired claim. $\qquad\square$

### A.5 Proofs for Section D

*Proof of Proposition 19.* Note that Theorem 4 of Immonen et al. (2023) showed that RePHINE is a graph isomorphism invariant. Since $\mathrm{RePHINE}^{mSpec}$ is a special case of RePHINE, it is also a graph isomorphism invariant. Theorem 2 of Ballester & Rieck (2024) shows that PH is a graph isomorphism invariant.

Now for EC, we observe that for a graph $G$ with pairs $(f_v, f_e)$, we can write

$$\chi(G_t^{f_v}) = \#\{(b_v, d_v) \in \mathrm{PH}^0(f) : b_v \leq t, d_v > t\} - \#\{(b_e, d_e) \in \mathrm{PH}^0(f) : b_e \leq t, d_e > t\}.$$

Here $(b_v, d_v)$ indicates that the pair is from 0-dimensional persistence, and $(b_e, d_e)$ indicates that the pair is from 1-dimensional persistence. We can similarly write $\chi(G_t^{f_e})$ using $\mathrm{PH}^1(f)$. Since PH is a graph isomorphism invariant and determines EC, we have that EC is a graph isomorphism invariant. Now, $\mathrm{EC}^m$ is a special case of EC, so it is also a graph isomorphism invariant.

Finally, if $\mathrm{RePHINE}^{Spec}$ is a graph isomorphism invariant, then so is $\mathrm{RePHINE}^{mSpec}$. It then suffices for us to check this for $\mathrm{RePHINE}^{Spec}$. We will first show that for $\mathrm{RePHINE}^{Spec}(f)^0$. Here we follow the suggestions laid out in Theorem 4 of Immonen et al. (2023) and decompose it in two steps.

1. From Theorem 4 of Immonen et al. (2023), we know that the original RePHINE with $(b, d, \alpha, \gamma)$ is an isomorphism invariant.

2. We also want to show that the map $G \mapsto \{\rho(C(v, d(v)))\}_{v \in V(G)}$ is an isomorphism invariant. Here, $\rho$ produces the spectrum of the connected component $v$ is in at its time $d(v)$. Since $\rho$ itself does not depend on choice, it suffices for us to show the following map is an isomorphism invariant:
$$G \mapsto \{C(v, d(v))\}_{v \in V(G)},$$
where $C(v, d)$ is the component $v$ is in at time $d$. The ambiguity comes in, depending on our choice, a vertex may very well die at different times.

   Now from the proof of Theorem 4 of Immonen et al. (2023), we already know that the multi-set of death times is an isomorphism invariant. Now a vertex death can only occur during a merging of old connected components $T_1, ..., T_n$ (with representatives $v_1, ..., v_n$) to a component $C$. Now under the RePHINE scheme, there is a specific procedure to choose which vertex. However, we see that any choice of the vertex does not affect the connected component that will be produced after merging. Thus, we will always be adding a constant $n - 1$ copies of $C$ to the function we are producing.

   For the real holes, we know from the proof of Theorem 4 of Immonen et al. (2023) that, it does not matter how each of the remaining vertices is matched to the real holes, since the rest of the vertices are associated in an invariant way. Hence, the production of graph Laplacians for the real holes will not be affected. Finally, the description above also shows that we can concatenate the pair $(b, d, \alpha, \gamma)$ with $(\rho)$ in a consistent way.

3. Hence, we see that $\mathrm{RePHINE}^{Spec}(f)^0$ is an isomorphism invariant.

Now for RePHINE$^{Spec}(f)^1$, the argument follows similarly as above. It suffices for us to check that the list $\{C(e, b(e))\}_{e \in \text{cycle}}$ is consistent. We observe that the birth of a cycle can only happen when an edge occurs that goes from a connected component $C$ back to itself. The only possible ambiguity is that, at the birth time, multiple edges are spawned at the same time, and an edge may or may not create a cycle based on the order it is added to the graph. However, regardless of the order, the resulting connected component that the edge belongs in will be the same. Thus, the list $\{C(e, b(e))\}_{e \in \text{cycle}}$ will be consistent. $\square$

### A.6  PROOFS FOR SECTION 6

*Proof of Proposition 6.* The intuitive idea is that the zeroth homology only captures the notion of connected components, and adding more edges between two nodes that are already connected does not affect this connectivity. To be more rigorous, we will invoke Theorem 1 and Theorem 2 of Immonen et al. (2023) to prove the first and second statements respectively. Although both theorems were proven for the case of simple graphs, their proofs easily generalize to graphs with multi-edges such that every edge between two fixed nodes has the same color. Hence, we can apply them to $M(c)$ and $M(t)$. In particular, we can see that there are no color-separating sets and no color-disconnecting sets between $M(c)$ and $M(t)$, so it follows that the 0-th dimensional persistence diagrams of $M(t)$ will be the same as that of $M(c)$. $\square$

*Proof of Proposition 7.* For ease of notation, for a graph representing a molecule, we write $G_1$ as the number of edges in $G$ labeled "single bond", $G_2$ as the number of edges in $G$ labeled "double bond", and $G_3$ as the number of edges in $G$ labeled "triple bond."

Although we only proved Theorem 3 for simple graphs, the proof of Theorem 3 to the bond counting representation of molecules we are considering here. In particular, the characterization of EC tells us that the EC diagram of $M(t)$ and $N(t)$ with respect to all possible $f_e$ are the same if and only if $M(t)_1 = N(t)_1, M(t)_2 = N(t)_2, M(t)_3 = N(t)_3$. Now $M(t)_1 = M(c)_1, M(t)_2 = 2M(c)_2$, and $M(t)_3 = 3M(c)_3$ (and similarly for $N$), so equivalently we have that

$$M(c)_1 = N(c)_1, M(c)_2 = N(c)_2, M(c)_3 = N(c)_3.$$

By Theorem 3, the equalities above are equivalent to the observation that the EC diagram of $M(c)$ and $N(c)$ with respect to all possible $f_e$ are the same. This proves the proposition. $\square$

## B  EXPRESSIVITY OF SPECTRAL INFORMATION

Suppose $K$ is an $n$-dimensional finite simplicial complex. There is a standard simplicial chain complex of the form

$$\ldots \xrightarrow{\partial_{n+1}} 0 \xrightarrow{\partial_{n+1}} C_n(K) \xrightarrow{\partial_n} \ldots \longrightarrow C_1(K) \xrightarrow{\partial_1} C_0(K) \xrightarrow{\partial_0} 0 \ .$$

Here each $C_i(K)$ has a formal basis being the finite set of $i$-simplicies in $K$, hence there is a way well-defined notion of an adjoint (which is the transpose) $\partial_i^T$ for each $\partial_i$. The $i$-th combinatorial Laplacian of $K$ is defined as

$$\Delta_i(K) = \partial_i^T \circ \partial_i + \partial_{i+1} \circ \partial_{i+1}^T.$$

$\Delta_i$ is a linear operator on $C_i(K)$. Note that when $K$ is a graph and $i = 0$, $\Delta_0(K)$ is exactly the graph Laplacian of $K$. It is a general fact that the dimension of $\ker \Delta_i(K)$ is the same as the $i$-th Betti number of $K$. Hence, the multiplicity of the zero eigenvalues of $\Delta_i(K)$ corresponds to the $i$-th Betti number of $K$. However, the Betti numbers of $K$ (harmonic information) do not give any information on the non-zero eigenvalues of $\Delta_i(K)$ (which we can think of as the non-harmonic information). This is the data that we would like to keep track of.

For a filtration of a simplicial complex $K$ by $\emptyset = K_0 \subseteq K_1 \subseteq \ldots \subseteq K_m = K$, Wang et al. (2020) proposed a persistent version of combinatorial Laplacians as follows.

**Definition 13.** *Let $C_q^t = C_q(K_t)$ denote the $q$-th simplicial chain group of $K_t$, $\partial_q^t : C_q(K_t) \rightarrow C_{q-1}(K_t)$ be the boundary map on the simplicial subcomplex $K_t$. For $p > 0$, we use $\mathbb{C}_q^{t+p}$ to denote the subset of $C_q^{t+p}$ whose boundary is in $C_{q-1}^t$ (in other words $\mathbb{C}_q^{t+p} := \{\alpha \in C_q^{t+p} \mid \partial_q^{t+p}(\alpha) \in C_{q-1}^t\}$).*

*We define the operator $\eth_q^{t+p} : \mathbb{C}_q^{t+p} \rightarrow C_{q-1}^t$ as the restriction of $\partial_q^{t+p}$ to $\mathbb{C}_q^{t+p}$. From here, we define the $p$-persistent $q$-combinatorial Laplacian $\Delta_q^{t+p}(K) : C_q(K_t) \rightarrow C_q(K_t)$ as*

$$\Delta_q^{t+p}(K) = \eth_{q+1}^{t+p}(\eth_{q+1}^{t+p})^T + (\partial_q^t)^T \partial_q^t.$$

Note that the multiplicity of the zero eigenvalues in $\Delta_q^{t+p}(K)$ coincides with the $p$-persistent $q$-th Betti number.

Now we will focus on the special case where $K = G$ is a graph. In this case, we only need to look at the $p$-persistent 1-combinatorial Laplacians and the $p$-persistent 0-combinatorial Laplacians. Our goal is to augment the RePHINE diagram, so we intuitively would like to include all the non-zero eigenvalues of the $p$-persistent $q$-combinatorial Laplacians in our augmentation. In this section, we will show that augmentation is no more expressive than simply focusing on the spectral information of the ordinary graph Laplacian.

**Lemma 1.** *On a graph $G$, the multi-set of non-zero eigenvalues of $\Delta_1(G)$ is the same as the non-zero eigenvalues of $\Delta_0(G)$.*

*Proof.* For ease of notation, we omit the parameter $G$ in the combinatorial Laplacian. Since $G$ has dimension 1, $\partial_0$ and $\partial_2$ are both 0. Hence, the two combinatorial Laplacians may be written as

$$\Delta_0 = \partial_1 \circ \partial_1^T \text{ and } \Delta_1 = \partial_1^T \circ \partial_1.$$

Let $v$ be an eigenvector of $\Delta_0$ corresponding to a non-zero eigenvalue $\lambda$, then

$$\Delta_1(\partial_1^T v) = \partial_1^T(\partial_1 \circ \partial_1^T(v)) = \partial_1^T(\Delta_0(v)) = \partial_1^T(\lambda v) = \lambda(\partial_1^T v).$$

Hence, $\partial_1^T v$ is an eigenvector of $\Delta_1$ with eigenvalue $\lambda$.

We also need to check that if $v, w$ are linearly independent eigenvectors of $\Delta_0$ with the same eigenvalue $\lambda$, then $\partial_1^T v$ and $\partial_1^T w$ are linearly independent. Suppose for contradiction this is not the case, then there exist coefficients $a, b \in \mathbb{R}$ (not all zero) such that

$$0 = a\partial_1^T v + b\partial_1^T w = \partial_1^T(av + bw).$$

This means that $av + bw \in \ker(\partial_1^T) \subset \ker(\Delta_0)$ is a non-zero eigenvector corresponding to the eigenvalue 0. However, we also know that $av + bw$ is a non-zero eigenvector of $\Delta_0$ corresponding to the eigenvalue $\lambda$. Thus, it has to be the case that $av + bw = 0$, so we have a contradiction.

Hence, the non-zero eigenvalues of $\Delta_0$ form a sub-multiset of that of $\Delta_1$. The other direction may also be proven using linear algebra. Alternatively, however, we observe that by the equality of the Euler characteristic,

$$|V(G)| - |E(G)| = \chi(G) = \dim\ker(\Delta_0) - \dim\ker(\Delta_1).$$

Rearranging the terms gives us

$$|V(G)| - \dim\ker(\Delta_0) = |E(G)| - \dim\ker(\Delta_1).$$

This means that $\Delta_1$ and $\Delta_0$ have the same number of non-zero eigenvalues, so their respective multi-sets of non-zero eigenvalues are equal. $\qquad\square$

Let $G$ be a graph and

$$\emptyset = G_0 \subseteq G_1 \subseteq G_2 \subseteq ... \subseteq G_m = G$$

be a sequence of subgraphs of $G$. Recall that $\Delta_q^{t+p}(G)$ denotes the $p$-persistent $q$-combinatorial Laplacian operator. We will first examine what happens when $q = 1$.

**Lemma 2.** *The 1-combinatorial $p$-persistence Laplacian $\Delta_1^{t+p}(G)$ is equal to $\Delta_1^t(G) = \Delta_1(G_t)$. Moreover, the non-zero eigenvalues of $p$-persistence $\Delta_1^{t+p}(G)$ are the same as the non-zero eigenvalues of $\Delta_0^t(G)$, accounting multiplicity.*

*Proof.* Recall that $\Delta_q^{t+p}(G)$ is defined as

$$\Delta_q^{t+p}(G) = \eth_{q+1}^{t+p}(\eth_{q+1}^{t+p})^T + (\partial_q^t)^T\partial_q^t.$$

When $q = 1$, we know that $\eth_2^{t+p}(G)$ is the zero matrix since $G$ is a graph, and hence

$$\Delta_1^{t+p}(G) = (\partial_1^t)^T\partial_1^t.$$

This is independent of $p$ and is just $\Delta_1^t(G)$. Finally, from Lemma 1, we have that $\Delta_1^t(G)$ and $\Delta_0^t(G)$ have the same multi-set of non-zero eigenvalues. $\qquad\square$

**Remark 2.** *This is reflective of the definition of the $p$-persistent $k$-th homology group of $G^t$, which is given by*

$$H_k^p(G^t) = \ker\partial_k(G^t)/(\operatorname{im}\partial_{k+1}(G^{t+p}) \cap \ker(\partial_k(G^t)))$$

*In this case, when $k = 1$, $\partial_{k+1}(G^{t+p}) = \partial_2(G^{t+p})$ is the zero map, so $H_1^p(G^t) = \ker\partial_1(G^t) = H^1(G^t)$. Hence, the $p$-persistent 1st homology groups of $G^t$ stays constant as $p$ varies. This is also reflective of the fact that an inclusion of subgraph $i : G \to G'$ induces an injective homomorphism $i_* : H_1(G) \to H_1(G')$.*

The focus of persistent spectral theory should then be on the data given by the graph Laplacians, so it makes sense for us to interpret what exactly $\Delta_0^{t+p}(G)$ is.

**Lemma 3.** *The $p$-persistent 0-combinatorial Laplacian operator of $G$*

$$\Delta_0^{t+p}(G) = \eth_1^{t+p}(\eth_1^{t+p})^T$$

*is the graph Laplacian of the subgraph of $G_{t+q}$ with all the vertices in $G_t$.*

*Proof.* Recall that $\mathbb{C}_0^{t+p} = \{\alpha \in C_q^{t+p} \mid \partial_1^{t+p}(\alpha) \in C_0^t\}$. This represents all the 1-simplicies (edges) in $G_{t+p}$ whose vertices are in $G_t$. The map $\eth_1^{t+p} : \mathbb{C}_1^{t+p} \to C_0^t$ is the restriction map on $\partial^{t+p}$ onto $\mathbb{C}_0^{t+p}$. Let $G'$ denote the subgraph of $G_{t+q}$ generated by vertices in $G_t$, then there are two vertical isomorphisms, by quite literally the identity map, such that the following diagram commutes,

$$
\begin{array}{ccc}
C_1(G') & \xrightarrow{\;\partial\;} & C_0(G') \\
\downarrow & & \downarrow \\
\mathbb{C}_1^{t+p} & \xrightarrow{\;\eth\;} & C_0^t
\end{array}
.
$$

Hence, the graph Laplacian of $G'$ is the same as the Laplacian $\Delta_0^{t+p}$. $\qquad\square$

**Corollary 2.** *Lemma 2 asserts that the non-zero eigenvalues of $\Delta_1^{t+p}(G)$ are the same as the non-zero eigenvalues of $\Delta_0^t(G) = \Delta_0(G_t)$. In the special case where we focus on filtrations of $G$ given by $(F_v, F_e)$ outlined in Section 2, we furthermore have that:*

1. *In the vertex filtration given by $F_v$, $\Delta_0^{t+p}(G)$ is the same as $\Delta_0(G_t)$.*
2. *In the edge filtration given by $F_e$, $\Delta_0^{t+p}(G)$ is the same as $\Delta_0(G_{t+p})$.*

*Proof.* For the vertex filtration, Lemma 3 implies that $\Delta_0^{t+p}(G)$ is the graph Laplacian of the subgraph of $G_{t+p}$ generated by the vertices in $G_t$. In the vertex filtration defined in Section 2, there are no additional edges added between the vertices in $G_t$ after time $t$, so the subgraph is the same as $G_t$.

For the edge filtration, Lemma 3 implies that $\Delta_0^{t+p}(G)$ is the graph Laplacian of the subgraph of $G_{t+p}$ generated by the vertices in $G_t$. However, all vertices are spawned at the start, so they have the same vertex set and the subgraph is just the entire graph $G_{t+p}$. $\square$

In particular, the non-zero eigenvalues of $p$-persistent $q$-combinational Laplacians for the edge filtration have the same expressive power of just including the non-zero eigenvalues of graph Laplacians at all time steps. Since the graph only changes whenever there is a death of a connected component or the birth of a cycle, it suffices for us to include the graph Laplacians at all cycle birth times and vertex death times, hence the construction in Definition 6.

## C  EXPRESSIVITY OF EC ON SIMPLICIAL COMPLEXES

Let $K$ be a simplicial complex of dimension $n$ where each vertex is assigned a color in some coloring set $X$. The Euler characteristic of $K$ is defined by

$$\chi(K) = \sum_{i=0}^{n} (-1)^n \#\{\text{simplicies in } K \text{ of dimension } i\}.$$

**Notation:** Let $c_1, ..., c_i \in X$ be a list of distinct colors with $i \leq j$, we define $S_K^j(c_1, ..., c_i)$ as the number of $j$-simplices in $K$ whose vertice's set of colors is $\{c_1, ..., c_i\}$.

Under suitable extension of the definition of EC and max EC diagrams to a simplicial complex, we can, in fact, characterize them completely in terms of the combinatorial data provided by $S_K^j(c_1, ..., c_i)$. We will make these definitions precise and in particular prove the following theorem.

> **Theorem 6.** *Let $K, M$ be simplicial complexes of dimension $n$. The following are equivalent:*
>
> 1. $\text{EC}(K, f_0, ..., f_n) = \text{EC}(M, f_0, ..., f_n)$ *for all possible $f_0, ..., f_n$. Here, each $f_i$ is an $i$-simplex coloring function, to be elaborated upon in Appendix C.*
> 2. $\text{EC}^m(K, f_0) = \text{EC}^m(M, f_0)$ *for all possible $f_0$.*
> 3. *For all $j \leq n$, $S_K^j(c_1, ..., c_i) = S_M^j(c_1, ..., c_i)$ for all distinct colors $c_1, ..., c_i$ with $1 \leq i \leq j$.*

We first extend our definition of EC and $\text{EC}^m$ to a simplicial complex $K$ of dimension $n$ with a vertex coloring function $c : V(K) \to X$.

**Definition 14.** *Let $K$ be a simplicial complex of dimension $n$ with vertex color set $X$. For $0 < i \leq n$, we define a color value function $f_i : \prod_{j=1}^{i+1} X \to \mathbb{R}_{>0}$ as follows:*

1. *For any permutation $\sigma \in S_{i+1}$, $f_i(\sigma(\vec{x})) = f_i(\vec{x})$.*

2. *For any $\vec{x}, \vec{y} \in \prod_{j=1}^{i+1} X$ such that $\vec{x}$ and $\vec{y}$ have the same coordinates modulo order and multiplicity, $f_i(\vec{x}) = f_i(\vec{y})$. For example, if $X = \{red, blue\}$, $f_2(red, red, blue) = f_2(blue, blue, red)$. The intuition is, as sets (so we forget about order and multiplicity), $\{red, red, blue\}$ is the same set as $\{blue, blue, red\}$.*

*We define the $i$-th simplex filtration function $f_i$ of $K$ as follows:*

- *For all simplices $\sigma$ with dimension less than $i$, $F_i(\sigma) = 0$.*

- *For all $i$-simplices $\sigma$ with vertices $v_0, ..., v_i$, $F_i(\sigma) := f_i(c(v_0), ..., c(v_i))$.*

- *For all simplices $\sigma$ with dimension greater than $i$, $F_i(\sigma) := \max_{i\text{-simplex } \tau \subset \sigma} f_i(\tau)$.*

*Note that when $K = G$ is a graph $f_0, f_1, F_0, F_1$ agrees with our construction of $f_v, f_e, F_v, F_e$. Finally, for each $i$ and $t \in \mathbb{R}$, we define*

$$K_t^{f_i} := F_i^{-1}((-\infty, t]).$$

From here, we can construct the EC and max EC diagram on simplicial complexes as follows.

**Definition 15.** *Let $K$ be a simplicial complex of dimension $n$ with simplex filtration functions $f_0, f_1, ..., f_n$. The EC diagram $\text{EC}(K, f_0, ..., f_n)$ of $K$ is composed of $n + 1$ lists $\text{EC}(K, f_i)$ for $i = 0, ..., n + 1$. For each $i$, write $a_1^i < ... < a_{n_i}^i$ as the list of values $f_i$ can produce, $\text{EC}(K, f_i)$ is the list $\{\chi(K_{a_j^i}^{f_i})\}_{j=1}^{n_i}$.*

*Suppose we are only given a vertex filtration function $f_0$. We define the max EC diagram of $K$ as $\text{EC}^m(K, f_0) = \text{EC}(K, f_0, g_1, ..., g_n)$. Here $g_i(\sigma) = \max_{i\text{-simplex } \tau \subset \sigma} \max_{v_i \in \tau} f_0(v_i)$.*

One can check that the $g_i$'s defining max EC are consistent with the setup in Definition 14. As an immediate corollary of Theorem 6, we will see that EC and $\text{EC}^m$ have the same expressive power on the level of simplicial complexes.

**Theorem 7.** EC *and* $\text{EC}^m$ *have the same expressive power.*

**Remark 3.** *The choice of Condition (2) in Definition 14 is intentional so that Theorem 7 would hold. The intuition is that Condition (2) corresponds to coloring the $i$-th simplex $\sigma$, with vertices $v_0, ..., v_{i+1}$, of $K$ with the feature being the set $\{v_0, ..., v_{i+1}\}$ that forgets about multiplicity and order. However, Theorem 7 is not true if we modify Condition (2) such that we color $\sigma$ with the feature being the multi-set $\{c(v_0), ..., c(v_{i+1})\}$ to remember multiplicity.*

We will now prove Theorem 6. To do this, we will first prove a lemma.

**Lemma 4.** *Let $K$ and $M$ be two simplicial complexes of dimension $n$, if they have the same max EC diagram for all possible $f_0$, then $K$ and $M$ have the same $f$-vector, meaning that their respective number of simplicies are the same at each dimension.*

*Proof.* For this proof, let $K^i$ denote the subcomplex of $K$ with all simplicies with dimension $\leq i$. By comparing any complete filtration, we will have that $\chi(K) = \chi(M)$ (so $\chi(K^{n-1}) = \chi(M^{n-1})$). Comnparing time $t = 0$ for any induced $i$-simplex coloring filtration by an injective $f_0$ will give us that $\chi(K^i) = \chi(M^i)$ for all $i = 0, ..., n-1$. From here we can see that they have the same $f$-vector. $\qquad\square$

Now we will finally prove Theorem 6.

*Proof of Theorem 6.* Clearly (1) implies (2). To see that (3) implies (1), we first note that (3) implies $K$ and $M$ have the same $f$-vector, so the starting values of their Euler characteristics for any $f_i$ are always the same. For any change of the Euler characteristic as time varies, we will be modifying the value by adding or subtracting values of the form $S_K^j(c_1, ..., c_i) = S_M^j(c_1, ..., c_i)$. Hence, their Euler characteristics will agree as time varies.

For (2) implies (3), we first note that Lemma 4 implies $K$ and $M$ have the same $f$-vector. We will now prove the claim from a backward induction on $j = 0, 1, ..., n$. Indeed, when $j = n$, we have that

1. We will induct on $1 \leq i \leq n$. For $i = 1$, we wish to show that $S_K^n(c_1) = S_M^n(c_1)$ for all $c_1 \in C$. Note that $S_K^n(c_1)$ is just the number of $n$-simplices whose vertices all have color $c_1$. Choose $f_0$ such that $c_1$ has the minimum value, then under the induced max $n$-simplex coloring filtration, we have that

$$\chi(K_{f_0(c_1)}^{f_n}) = \chi(M_{f_0(c_1)}^{f_n})$$

   Since $K$ and $M$ have the same $f$-vector by Lemma 4, we conclude that $S_K^n(c_1) = S_M^n(c_1)$.

2. Suppose this is true up to $1 \leq i = k < n$. We wish to show this is true for $i = k + 1$. Indeed, let $c_1, ..., c_{k+1}$ be $k + 1$ distinct colors. Choose $f_0$ such that $c_1$ has the minimum value, $c_2$ is the second smallest, and so on until $c_{k+1}$. Under the induced max $n$-simplex coloring filtration, we have that

$$\chi(K_{f_0(c_{k+1})}^{f_n}) = \chi(M_{f_0(c_{k+1})}^{f_n})$$

   Subtracting their Euler characteristics at the time step $f_0(c_k)$, we have that

$$(-1)^n \sum_{a \in \mathcal{A}} S_K^n(a) = (-1)^n \sum_{a \in \mathcal{A}} S_M^n(a).$$

   where $\mathcal{A}$ is the collection of subsets of $\{c_1, ..., c_{k+1}\}$ that contains $c_{k+1}$. Since the inductive hypothesis is true up to $i = k$, we know that for all $a \in \mathcal{A}$ such that $|a| \leq k$, $S_K^n(a) = S_M^n(a)$. Hence cancelling both sides gives us

$$\sum_{a \in \mathcal{A}, |a| > k} S_K^n(a) = \sum_{a \in \mathcal{A}, |a| > k} S_M^n(a).$$

Now, there is only one element in $\mathcal{A}$ with $|a| > k$, namely $a = \{c_1, ..., c_{k+1}\}$. Hence, we have proven this for the case $i = k + 1$.

3. Thus, by induction, we have shown this for $j = n$.

Now we will induct down from $j = n$. Indeed, suppose this is true up to $0 < j = k + 1 \le n$, we wish to show this is true for $j = k$. Now we wish to show that $S_K^k(c_1, ..., c_i) = S_M^k(c_1, ..., c_i)$ for all distinct colors $c_1, ..., c_i$ and $1 \le i \le k$. We will do this by an induction on $i$.

1. For $i = 1$, we wish to show that $S_K^k(c_1) = S_M^k(c_1)$. Indeed, choose $f_0$ such that $c_1$ has the minimum value. Then we have that

$$\chi(K_{f_0(c_1)}^{f_k}) = \chi(M_{f_0(c_1)}^{f_k})$$

Since $K$ and $M$ have the same $f$-vector, cancelling that out gives us that

$$\sum_{\ell=k}^{n} S_K^\ell(c_1) = \sum_{\ell=k}^{n} S_M^\ell(c_1).$$

By our inductive hypotehsis on $j$, we know that $S_K^\ell(c_1) = S_M^\ell(c_1)$ for all $\ell > k$, hence subtracting them off gives us that $S_K^k(c_1) = S_M^k(c_1)$.

2. Suppose this is true up to $1 \le i = \ell < k$. We wish to show this is true for $i = \ell + 1$. Indeed, let $c_1, ..., c_{\ell+1}$ be $\ell + 1$ distinct colors. Choose $f_0$ such that $c_1$ has the minimum value, $c_2$ is the second smallest, and so on until $c_{\ell+1}$. Under the induced max $k$-simplex coloring filtration, we have that

$$\chi(K_{f_0(c_{\ell+1})}^{f_k}) = \chi(M_{f_0(c_{\ell+1})}^{f_k})$$

Since they have the same Euler characteristics at the time step $f_0(c_{\ell+1})$, we can cancel those terms out and obtain

$$\sum_{a \in \mathcal{A}} \sum_{p=k}^{n} (-1)^p S_K^p(a) = \sum_{a \in \mathcal{A}} \sum_{p=k}^{n} (-1)^p S_M^p(a).$$

Here $\mathcal{A}$ is the collection of all subsets of $\{c_1, ..., c_{\ell+1}\}$ that contains the color $c_{\ell+1}$. Now by the inductive hypothesis on $j$, we know that $S_K^p(a) = S_M^p(a)$ for all $p > k$, so we can cancel the expressions and obtain

$$(-1)^p \sum_{a \in \mathcal{A}} S_K^k(a) = (-1)^p \sum_{a \in \mathcal{A}} S_M^k(a).$$

Now by the inductive hypothesis for $i$, we have that for all $|a| < \ell + 1$, $S_M^k(a) = S_K^k(a)$, hence we have that

$$(-1)^p \sum_{a \in \mathcal{A}, |a| > \ell} S_K^k(a) = (-1)^p \sum_{a \in \mathcal{A}, |a| > \ell} S_M^k(a).$$

There is only one subset of size $\ell + 1$, namely $\{c_1, ..., c_{\ell+1}\}$. Hence, we conclude that $S_K^k(c_1, ..., c_{\ell+1}) = S_M^k(c_1, ..., c_{\ell+1})$.

Thus, by the principle of induction, we have proven that (2) implies (3). $\qquad\square$

# D   WELL-DEFINEDNESS AND DEATH-TIME FILTRATION

We will verify in Appendix A.5 that all the diagrams in Section 2 are graph isomorphism invariants.

**Proposition 19.** *The diagrams* PH, RePHINE, RePHINE$^m$, EC, EC$^m$, RePHINE$^{Spec}$, *and* RePHINE$^{mSpec}$ *are all well-defined graph isomorphism invariants.*

In the construction of max RePHINE in Definition 4, we considered an edge filtration of $G$ whose values are dependent on the color values of the vertices in $G$ under $f_v$. For a vertex $w \in V(G)$, $f_v(w)$ is the birth time of the vertex in the filtration given by $f_v$. Therefore, the edge filtration in the max RePHINE diagram may be thought of as a certain "**birth-time filtration**".

As a dual notion, this motivates the idea for us to consider a "**death-time filtration**". A possible proposal one might reasonably consider is as follows.

**Proposal 1.** *Let $f_v : X \to \mathbb{R}$ be a vertex function such that $f_v > 0$. We use this to define an edge filtration of $G$ as follows. At time $t = 0$, $G_0$ is composed of all the vertices of $G$. At finite time $t > 0$, $G_t = G_0 \cup \{(v_0, v_1) \in E(G) \mid \max(d(v_0), d(v_1)) \leq t\}$. Here $d$ denotes the death time of the vertex in the filtration $\{G_t^{f_v}\}_{t \in \mathbb{R}}$. At time $t = \infty$, we add the remaining edges back in.*

This may appear to be a reasonable definition, as we have simply replaced the birth time of the vertex with the death time of the vertex. However, while there is a consistent function (ie. $f_v$) that assigns a vertex $v$ to its birth time, the death time of a vertex $v$ is not a function and can depend on choices made during the vertex filtration. We illustrate this problem with the following example.

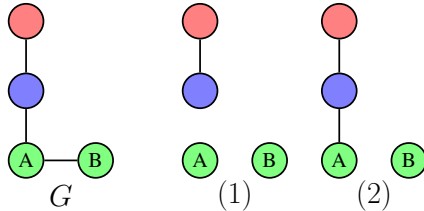

Figure 5: Illustration of a graph $G$ showing the proposed death time filtration is inconsistent if $f_v$ assigns 'green" to the lowest color value. (1) represents the filtration if we choose $v_2$ to be the real hole for time $t \to \infty$. (2) represents the filtration if we choose $v_3$ to be the real hole for time $t \to \infty$.

**Example 2.** *Let $G$ be the graph indicated in Figure 5 with vertex color function $f_v$ given by*

$$0 < f_v(\text{green}) = 1 < f_v(\text{blue}) = 2 < f_v(\text{red}) = 3.$$

*The vertex filtration yields the persistence diagram $\{(1, \infty), (1, 1), (2, 2), (3, 3)\}$. The ambiguity comes whether $A$ or $B$ is the real hole. If the vertex $A$ is the real hole, then the corresponding "death time filtration" when $t >> 3$ would be the graph labeled (1) in Figure 5, which has 3 connected components. If the vertex $B$ is the real hole, then the corresponding "death time filtration" when $t >> 3$ would be the graph labeled (2) in Figure 5, which has 2 connected components.*