# OpenReview forum: "Topological Data Analysis on Graphs: Euler characteristics, Persistent Homology, or Spectrum?"
_ICLR.cc/2025/Conference — Submitted to ICLR 2025_

### Official Review · Reviewer_icUB · 2024-10-25

**Soundness:** 2
**Presentation:** 1
**Contribution:** 2
**Rating:** 3
**Confidence:** 5

**Summary:**

The paper establishes several results on topological descriptors on graph representation learning.

**Strengths:**

The paper studies three topological descriptors—Euler Characteristics, Persistent Homology, and Laplacian Spectra—from various perspectives and compares their features and effectiveness.

**Weaknesses:**

I found the paper difficult to follow due to a lack of cohesion among the presented results. The paper includes multiple, somewhat incremental findings that are loosely grouped under a single theme but lack a clear, unifying narrative. Each topic examined by the authors requires a much deeper, individual analysis; however, the paper only provides limited advancements in each area. Moreover, these results appear largely unrelated to one another, creating an impression of a collection of disjointed findings rather than a focused contribution to the field of topological descriptors in graph representation learning. I will address some points in more detail below.

1. **Expressivity of PH or TD**: The expressivity of persistent homology (PH) or topological descriptors (TD) is difficult to define without specifying the space of filtration functions. Without this specification, one could theoretically achieve any level of distinguishing power between graphs by selecting suitable filtration functions (or colorings). Therefore, the current results lack meaning from a machine learning perspective, as they do not offer a robust comparison to graph neural network (GNN) expressivity.

2. **Stability**: The stability result presented here is incremental, as it mainly builds upon foundational work by Cohen-Steiner (2006). There is limited novel contribution to the understanding of stability in this context.

3. **Product of Graphs**: The purpose of this section is unclear, as it does not seem connected to the main theme of the paper. The motivation for ML and relevance to the other sections should be clarified.

4. **Death-time Filtration**: The relevance of this section to the other sections is also unclear. Its connection to the overall analysis needs to be better justified. The authors derive several new notions and make comparisons between them. However, the motivation is again unclear.

5. **Molecular Graphs**: Since this is an ML venue, I would expect experimental validation. Some experiments demonstrating the utility of the discussed approaches on molecular graphs would be valuable to illustrate practical implications.

**In summary**, the paper discusses several interesting topics; however, it lacks focus and depth. I recommend narrowing the scope to one or two of these topics, accompanied by a much deeper analysis, with clearly defined goals and statements. This approach would lead to a more substantial and coherent contribution.

**Questions:**

See weaknesses.

---

> ### Author Response · Authors · 2024-11-27
>
> We are grateful to Reviewer icUB for their time and detailed comments. We reply to your comments/questions below.
>
> ```
> The expressivity of persistent homology (PH) or topological descriptors (TD) is difficult to define without specifying the space of filtration functions. Without this specification, one could theoretically achieve any level of distinguishing power between graphs by selecting suitable filtration functions (or colorings).
> ```
> Thank you for the opportunity to clarify aspects of our work. In fact, we did clarify what the space of filtration functions are in Definition 1 (starting at Line 147) of the original submission. This is now Line 149 of the new submission. The color-based vertex and edge filtration functions on the graph $G$ are given by $F_v$ and $F_e$, which are induced from the functions $f_v: X \to \mathbb{R}$ and $f_e: X \times X \to \mathbb{R}_{>0}$.
>
> ```
> Stability: The stability result presented here is incremental, as it mainly builds upon foundational work by Cohen-Steiner (2006). There is limited novel contribution to the understanding of stability in this context.
> ```
>
> Thank you for the comment! There are three difficulties in extending the result to RePHINE. The first was that the original definition of Bottleneck distance did not generalize to RePHINE diagrams as it did not take into account of the $\alpha$ and $\gamma$ parameters. The second is that the original proof for Bottleneck stability is not applicable to our setting because it does not account for the $\alpha$ and $\gamma$ parameters. While the $\alpha$ values can be interpreted as the birth time of the vertices, there was not a clear interpretation of what the $\gamma$ values are in terms of the birth time / death time of a simplex. Furthermore, it was unclear how the setting of Bottleneck stability would change since the original metric on persistence pairs in Cohen-Steiner et al. (2006) were given by the $\ell_{\infty}$-norms, but in our paper the norm we adopted is the $\ell_1$-norm.
>
> In the new version of the manuscript, we have also investigated and added in a discussion on the stability of $\operatorname{RePHINE}^{Spec}$. Specifically, we have proven that $\operatorname{RePHINE}^{Spec}$ is locally stable under the assumption that the functions $f_v: X \to \mathbb{R}$ (which assigns each vertex color type with a real value) and $f_e: X \times X/\sim \to \mathbb{R}_{>0}$ (which assigns each edge volor type with a positivie real value) are both injective.
>
> By **locally stable**, we mean that tiny variations in the filtration functions used will not drastically change the output of $\operatorname{RePHINE}^{Spec}$. In our original submission, what we proved for $\operatorname{RePHINE}$ is actually somewhat stronger - we showed that the bottleneck distance of any two RePHINE diagrams on the same graph are bounded in terms of some variant of the $\ell^{\infty}$ metric on the filtration functions that created them. This is a global statement of stability, whereas it suffices for local stability to hold in applications.
>
> While $\operatorname{RePHINE}$ was shown to be globally stable by us, we also showed that $\operatorname{RePHINE}^{Spec}$ is not globally stable. Furthermore, when injectivity does not hold, it may not be locally stable either (but on a measure zero subset of the space of filtration functions). We have provided counter-examples for this at the end of Appendix A.3 (Example 1).
>
> To discuss the word **local**, we needed a notion of topology on $(f_v, f_e)$. The convenient choice for us turned out to be thinking of $(f_v, f_e)$ as an element of $\mathbb{R}^{n} \times \mathbb{R}^m$ (where $n$ is the size of the vertex coloring set $X$ and $m$ is the size of the correspondent edge coloring set). In this case, $f_v$ being injective is the same as saying its coordinates have no repeated entries. The main idea behind proving the local stability of $\operatorname{RePHINE}^{Spec}$ was that the proof for the stability $\operatorname{RePHINE}$ may be adapted in a convex open neighborhood of $f_v$ in $\mathbb{R}^n$ and $f_e$ in $\mathbb{R}^m$.

---

> ### Author Response · Authors · 2024-11-27
>
> ```
> Product of Graphs: The purpose of this section is unclear, as it does not seem connected to the main theme of the paper. The motivation for ML and relevance to the other sections should be clarified.
> ```
> Thank you for giving us the opportunity to clarify our work! A key motivation for us to study graph products is that, when dealing with large or complex graphs that have the structural property of being some product, it is often easier to work with the components of the graph product rather than the graph as a whole. This decomposition can be particularly useful in how they relate to persistent homology methods and spectral methods. In particular, we were interested in how much topological information about the whole graph we can recover by just analyzing the components. This is for example the content of Proposition 3, 4, and 5 in the original manuscript. In Proposition 3, we show that the product of vertex filtrations is the max of vertex colors. In Proposition 4, the product of edge filtrations is the natural edge coloring on the product. These two propositions give a natural way to decompose two color-based filtrations of graph products in terms of its components. Finally, Proposition 5 gives a way to compute the 0-dimensional persistence diagram of the graph product based on just its components.
>
> We have also added a description of these motivation for graph products at the beginning of the section.
>
> ```
> Death-time Filtration: The relevance of this section to the other sections is also unclear. Its connection to the overall analysis needs to be better justified. The authors derive several new notions and make comparisons between them. However, the motivation is again unclear.
> ```
>
> Thank you for the opportunity to clarify aspects of our work. In fact, we did we explain what the motivation for death-time filtrations are and how they relate back to the overall analysis on line 454 - 459 of the original submission. As explained in the paper, in the construction of the max versions of $\operatorname{RePHINE}^{Spec}, \operatorname{RePHINE}, $ and $\operatorname{EC}$, for an edge $(w_1, w_2)$, the edge value $f_e(v, w)$ is defined as the minimum of $f_v(w_1)$ and $f_v(w_2)$. On the other hand, $f_v(w_1)$ and $f_v(w_2)$ are the birth times of the vertices $w_1$ and $w_2$ in the vertex filtration. Thus, a motivating question to ask is - **what if we replace the birth time in the definition of the max versions with the death time?**
>
> Based on your input, we have moved the discussion on the death-time filtrations to the appendix.
>
> ```
> Molecular Graphs: Since this is an ML venue, I would expect experimental validation. Some experiments demonstrating the utility of the discussed approaches on molecular graphs would be valuable to illustrate practical implications.
> ```
> We greatly appreciate your feedback. We focus here on several key theoretical underpinnings from multiple perspectives to bridge prominent gaps in our understanding of various TDA methods from a machine learning perspective.
>
> We fully agree with you that a comprehensive empirical investigation would be useful, and hope that the theoretical insights presented in this work pave way for such detailed empirical investigations in future studies.
>
> ```
> I recommend narrowing the scope to one or two of these topics, accompanied by a much deeper analysis, with clearly defined goals and statements.
> ```
> We greatly appreciate your perspective and feedback. The key reason we focus on all the three axes – expressivity, computation, and stability – is that they together provide a rather complete and more realistic picture of the applicability of the different methods than if considered in isolation.  For example, methods such as those based on Laplace spectrum and PH are more expressive than EC, but in practice, depending on the computational budget, the latter might still be preferable in some settings.
>
> -----
> Many thanks for your insightful review and constructive feedback. We hope your concerns have been satisfactorily addressed, and if so, would appreciate if you could revisit your score to reflect the same. We are also committed to engaging further if you have any additional questions, concerns, or suggestions.

---

> ### Comment · Reviewer_icUB · 2024-11-28
>
> Thank you for your responses. While I acknowledge that this work presents several interesting findings, I believe it would benefit from a major revision to narrow its scope and provide a deeper analysis of the key results. As such, I will keep my original score.

---

> > ### Author Response · Authors · 2024-11-29
> >
> > Thank you for your follow up comment! We want to emphasize that we did follow up on your suggestion of narrowing the scope of the paper and focusing on the key results in the updated version of the manuscript. In particular, we moved the discussion of death-time filtrations to the appendix so that the main text is concerned with fewer topics. We added a new result on the local stability of $\operatorname{RePHINE}^{Spec}$ and thereby dived deeper into the stability analysis. We also connected the motivations of graph products more closely to the theme of the paper.
> >
> > In light of these efforts, we hope you would reconsider raising your score for the review. Are there any further specific topics in this paper that you would you suggest us to further narrow our scope on?

---

> > ### Author Response · Authors · 2024-12-03
> > **Follow-up Comment**
> >
> > Dear Reviewer icUB,
> >
> > As the discussion period is ending soon, we wanted to check-in again. We would like to re-iterate that we have addressed many specific points you have raised in your initial comment. Notably,
> >
> > - **Expressivity of PH or TD**: We clarified that we did specify what the space of filtration function is in our original submission.
> > - **Stability**: We explained why extending the stability of PH to RePHINE was non-trivial. Furthermore, in our new response, we added a new result on the local stability of $\operatorname{RePHINE}^{Spec}$.
> > - **Product of Graphs**: We clarified the motivation for graph products and included them in the updated manuscript. To add to our previous comment, there are real datasets where graph products find use; e.g., the authors of [1] use  (a) the  COIL-20 dataset for multi-view object clustering, and (b) real air-quality spatiotemporal data with the objective of learning one graph factor for estimating dependencies between air quality measurements in different cities and another graph factor to model the seasonal variations. The common theme is that they show up naturally for multi-view and/or multi-domain data.
> > - **Death Time Filtration**: We clarified that we did specify what the motivations for the section were in our original submission.
> > - **Narrowing the Scope with Deeper Analysis**: We did follow the advice by moving the discussion of death-time filtrations to the appendix and expanded the discussion on the stability analysis.
> >
> > If you have any specific questions or concerns, we would be happy to address them. We would appreciate it if you could consider raising the score. Thank you again for your feedback and help to improve the paper.
> >
> > [1] Sai Kiran Kadambari and Sundeep Prabhakar Chepuri, **Product Graph Learning from Multi-domain Data with Sparsity and Rank Constraints**.

---

### Official Review · Reviewer_pi7v · 2024-11-02

**Soundness:** 2
**Presentation:** 2
**Contribution:** 2
**Rating:** 3
**Confidence:** 4

**Summary:**

This paper seeks to systematically study and provide advice (in the form of new contributions) on three topological descriptors for graph neural networks in terms of expressivity, stability, and computational cost.

**Strengths:**

The paper seeks to rigorously study and provide a convincing basis for the role of topology in graph neural networks, which I agree with the authors is not yet convincing.  I think the paper does a good job at outlining the challenges of topological methods in computational settings, especially in deep learning and graph neural networks.

**Weaknesses:**

Some of the main conclusions and findings of the paper are not that surprising, such as the Euler characteristic being strictly less expressive than persistent homology.  This is in quite fact obvious, given that the (persistent) Euler characteristic is an alternating sum of Betti numbers, while persistent homology by construction contains more information.  Overall, while I agree with the need for an unbiased and rigorous study of the effectiveness of topological methods in graph neural networks and I believe that this part of the study was carried out well, I am still not convinced by the place that topology has in graph neural networks and I am not convinced by the new contributions proposed in the paper, which in some cases, I see as straightforward extensions of existing methods.  Perhaps it is an artifact of needing to "sell" the new contribution.  I almost feel like the paper would have been stronger as a rigorous computational survey paper, systematically studying the strengths and limitations of topological methods, with a presentation of straightforward/obvious ways to fix the limitations.

The paper is also not very well written.  I am missing the traditional summarized bullet point list clearly listing all contributions of the work.

**Questions:**

- Do you mean Euler characteristic or persistent Euler characteristic?  Definition 5 is the static version, but a persistent version is cited with Turner et al. being the reference.
- Overall, I find the presentation of the topological descriptors studied confusing; it is not very clear to me whether the persistent version is being studied for the three types or the static ones, although the title of the paper and the introduction seems to emphasize persistent versions of all descriptors.  Why are persistent versions studied in particular and not static ones?
- Should the authors choose to study static topological descriptors as well, I would be interested in what can be said about the Hodge Laplacian?  The kernel is known to encode geometric information about the homology of simplicial complexes (and therefore graphs as a lower dimensional special case); other eigenspaces do the same for graphs.  However, for the persistent version (which appears to be the one studied in this paper – again, not always clear...), only the kernel has been studied.
- What can be said in addition to expressivity?  What about interpretability of topological methods?  For this point, I admit that my rating of "soundness" might be somewhat unfair, as I think this paper would be stronger as a review/experimental study paper (as mentioned in the "weaknesses" section) and I would be open to raising my score for this criterion.  Although it may be (substantially) more work, I still feel like it is a plausible question since in some instances, interpretability and expressivity can be seen as related, especially in considering topological notions.

---

> ### Author Response · Authors · 2024-11-27
>
> We are grateful to Reviewer pi7v for their time and detailed comments. We reply to your comments/questions below.
>
> ```
> Some of the main conclusions and findings of the paper are not that surprising, such as the Euler characteristic being strictly less expressive than persistent homology. This is in quite fact obvious, given that the (persistent) Euler characteristic is an alternating sum of Betti numbers, while persistent homology by construction contains more information.
> ```
> We agree with the reviewer. It is only for completeness that we concluded it in the statement of Theorem 1. The purpose of Theorem 1 (and its proof) is to give an organized summary of comparisons and a concrete record for a collection of (counter)examples to rigorously justify the distinctions. We believe that the most interesting parts of Theorem 1 come in the incomparability of $\operatorname{RePHINE}^{mSpec}$ and $\operatorname{RePHINE}$, and the equivalence of $\operatorname{EC}$ and $\operatorname{EC}^m$, which are non-trivial and require subtle arguments and insights. Furthermore, Theorem 1 is only one of the items on the table of contributions (Figure 1).
>
> ```
> Do you mean Euler characteristic or persistent Euler characteristic? Definition 5 is the static version, but a persistent version is cited with Turner et al. being the reference.
> ```
> We thank the reviewer for raising this question. Definition 5 is the persistent version of Euler characteristic we had in mind, where we did a simplification from the original setting of the Euler characteristic transform (ECT) in Turner et al. The ECT in Turner et al. originally is an integeral transform that takes in a shape $S$ embedded in Euclidean space $\mathbb{R}^n$ and produces a function
>     $$\mathbb{S}^{n-1} \times \mathbb{R} \to \mathbb{Z}, (\nu, t) \mapsto \chi(\{x \in S\ |\ x \cdot \nu \leq t\}).$$
>     When we adapt the definition to our setting, there are two changes we had to make. First, our graph is not embedded in some particular Euclidean space, so there is no clear notion of directional transform. Therefore, instead of considering the directions given by $\mathbb{S}^{n-1}$, our directions are the two color filtrations given by vertices and edges.
>
> Since there are only two filtrations in this case, and the Euler characteristic can only change whenever $f_v$ (resp. $f_e$) changes. Our ECT can be simplified as just two lists of integers, which is currently in Definition 5.
>
> We also omitted the time variable from the output because for two different graphs with the same $f_v$ and $f_e$ (ie. in expressivity comparisons), both graphs will experience possible changes in their EC at the exact same time-stamp. The inclusion of the time variable only becomes important in stability analysis (ie. different color filtrations on the same graph.)
>
>
> ```
> Overall, I find the presentation of the topological descriptors studied confusing; it is not very clear to me whether the persistent version is being studied for the three types or the static ones, although the title of the paper and the introduction seems to emphasize persistent versions of all descriptors. Why are persistent versions studied in particular and not static ones?
> ```
> We thank the reviewer for raising this question. As a general question, we want to ask if the reviewer could clarify what a **static** vs. **persistent** descriptor is in their mind. To us, whenever we are looking at how a topological invariant changes over a given filtration, we would consider that as ``persistent". Whereas to us, a **static** descriptor is defined on a graph without the usage of a filtration. Under our interpretation, all of our descriptors that show up in Section 2 are persistent versions.
>
> The reason why we want to study persistent versions instead of static versions is because, under our interpretation, capturing how topological data changes over time contains more information than looking at a graph without filtrations. For example, consider two graphs $G$ and $H$ where $G$ is the circle graph on $3$ vertices, all labeled red, and $H$ is the circle graph on $3$ vertices, with two labeled yellow and one labeled red. If we only look at the static Euler characteristic of both graphs, then under our interpretation they would both output $0$, so there is no difference. However, if we look at how the Euler characteristic changes with an induced color filtrations on $G$ and $H$, then their EC diagrams would be different
>
> We have added a clarification of that we mean **persistent** in Section 2 of the manuscript.

---

> > ### Author Response · Authors · 2024-11-27
> >
> > ```
> > Should the authors choose to study static topological descriptors as well, I would be interested in what can be said about the Hodge Laplacian? The kernel is known to encode geometric information about the homology of simplicial complexes (and therefore graphs as a lower dimensional special case); other eigenspaces do the same for graphs. However, for the persistent version (which appears to be the one studied in this paper – again, not always clear...), only the kernel has been studied.
> > ```
> > We thank the reviewer for the suggestion. The Hodge Laplacian is very interesting, and we will certainly consider them in future research. In the main body of the paper, we only considered the graph Laplacian. We did not only study its kernel, though, but also its complement. In Definition 6 of the paper, we included precisely only the non-zero eigenvalues of the graph Laplacian (so everything not in the kernel) in our definition of $\operatorname{RePHINE}^{Spec}$.
> >
> > In Appendix B, we did consider a ``persistent Laplacian" defined in Wang et al. (2020). However, we showed in the same appendix that keeping track of the persistent Laplacians of a color-based filtration in the context of graphs is no more as expressive as keeping track of how the non-zero eigenvalues of the ordinary graph Laplacian changes over the filtration.
> >
> > ```
> > What can be said in addition to expressivity? What about interpretability of topological methods? For this point, I admit that my rating of "soundness" might be somewhat unfair, as I think this paper would be stronger as a review/experimental study paper (as mentioned in the "weaknesses" section) and I would be open to raising my score for this criterion. Although it may be (substantially) more work, I still feel like it is a plausible question since in some instances, interpretability and expressivity can be seen as related, especially in considering topological notions.
> > ```
> > We thank the reviewer for the question and suggestion. For the interpretability of the Euler characteristics (Definition 5) specifically, we did give a completely combinatorial interpretation of the expressivity of EC diagrams in Theorem 3. In the original paper that proposed RePHINE (Immonen et al.), the authors also gave an interpretation of the persistent homology (PH) diagrams on vertex-level filtrations in terms of a combinatorial object they called **color-separating sets**. For PH diagrams on edge-level filtrations, they also gave an interpretation in terms of another combinatorial object they called **color-disconnecting sets**.
> >
> > -----
> >
> > Many thanks for your insightful review and constructive feedback. We hope your concerns have been satisfactorily addressed, and if so, would appreciate if you could revisit your score to reflect the same. We are also committed to engaging further if you have any additional questions, concerns, or suggestions.

---

> ### Author Response · Authors · 2024-12-03
> **Follow-up Comment on Rebuttal**
>
> Dear Reviewer pi7v,
>
> As the discussion period is ending soon, we wanted to check-in again. We have addressed most of your concerns in the rebuttal, notably:
>
> -  **Missing the list of all contributions**: Figure 1 of the current submission lists a full table summarizing our contributions in the paper.
>
> - **Static vs. Persistent**: We have clarified our interpretation of what persistent and static methods are, explained that all methods we consider are persistent in the document, and explained why persistent method is more advantageous than static.
>
> - We also wanted to bring your attention that we also analyzed and prove the (local) stability of $\operatorname{RePHINE}^{Spec}$ in the updated manuscript, which we believe adds more novelty to our submission. See our global response for more details.
>
> We hope your concerns have been satisfactorily addressed, and if so, would appreciate if you could revisit your score to reflect the same. If you have any specifc questions or concerns, we would be happy to address them. Thank you again for your feedback and help to improve the paper.

---

### Official Review · Reviewer_NVLY · 2024-11-03

**Soundness:** 2
**Presentation:** 1
**Contribution:** 2
**Rating:** 5
**Confidence:** 5

**Summary:**

The authors analyse three topological descriptors Persistent Homology, Euler Characteristic and Laplacian Spectrum from expressivity, computation and stability standpoint. The authors also introduce weaker but computationally more feasible topological descriptors. They show that $RePHINE^{spec}$ is among the most expressive topological descriptors. Further, they show that $RePHINE$ diagrams are stable and discuss how the analysis would generalize for product of graphs.

**Strengths:**

I like the problem that the authors are attempting to solve, which is to have a comprehensive guide about which topological descriptor to use on what kind of data/tasks. I feel that solving this problem would really help TDA in ML community.

**Weaknesses:**

Last line of the first paragraph “the theoretical foundations of TDs remain rather elusive.” This statement, as it reads, is incorrect. All the topological descriptors have a rich theory backing them. What the authors, perhaps, want to say is the expressivity analysis of TDs is rather elusive.

Definition 1: $f_e(c,c’) = f_e(c,c’)$ -> $f_e(c,c’) = f_e(c’,c)$

Most of the comparisons in Theorem 1 seem fairly obvious from the definitions.

I don’t understand the point of stating Theorem 2 separately. Isn’t it already stated in Theorem 1 that $EC$ and $EC^m$ have the same expressive power?

The flow seems to be lacking from Section 4 onwards. I don’t quite get the motivation behind the product of graphs and death-time filtrations.

Section 5 seems something new, although I am not sure about its applicability as such.

I didn’t quite understand the overall point being made by Section 7.

In Definition 8, what does $h_1 ~ h_2$ mean?

Line 491: Bond Coloring Representation -> Bond Counting Representation?

Overall, I think that the paper needs significant revisions before it is ready for acceptance.

**Questions:**

The stability section just talks about the stability of $RePHINE$ diagrams, which is a straightforward extension of the stability of persistence diagrams. What about the stability of $RePHINE^{spec}$? Can that be characterized as well? What about the stability of EC?

---

> ### Author Response · Authors · 2024-11-27
>
> We are grateful to Reviewer NVLY for their time and detailed comments. We reply to your comments/questions below.
>
>  ```
> Last line of the first paragraph “the theoretical foundations of TDs remain rather elusive.” This statement, as it reads, is incorrect. All the topological descriptors have a rich theory backing them. What the authors, perhaps, want to say is the expressivity analysis of TDs is rather elusive.
> ```
> We thank the reviewer for giving this suggestion. We have changed the wording of the sentence in the edited draft from **theoretical foundations** to **expressivity analysis**.
>
> ```
> Definition 1: $f_e(c,c’) = f_e(c,c’) -> f_e(c,c’) = f_e(c’,c)$
> ```
> We thank the reviewer for pointing out this typo, and we have corrected it in the edited draft.
>
> ```
> Most of the comparisons in Theorem 1 seem fairly obvious from the definitions.
> ```
> We thank the reviewer for raising this comment. We agree that most of the comparisons in Theorem 1 are straight-forward. The purpose of Theorem 1 (and its proof) is to give an organized summary of comparisons and a concrete record for a collection of (counter)examples to rigorously justify the distinctions. We believe that the most interesting parts of Theorem 1 come in the incomparability of $\operatorname{RePHINE}^{mSpec}$ and $\operatorname{RePHINE}$, and the equivalence of $\operatorname{EC}$ and $\operatorname{EC}^m$, which are non-trivial and require subtle arguments and insights.
>
> ```
> I don’t understand the point of stating Theorem 2 separately. Isn’t it already stated in Theorem 1 that $EC$ and $EC^m$ have the same expressive power?
> ```
> We thank the reviewer for raising this comment. We stated Theorem 2 apart from Theorem 1 to emphasize the importance/surprise that $\operatorname{EC}$ and $\operatorname{EC}^m$ are equivalent. Please let us know if you think it would help to remove it instead.
>
> ```
> The flow seems to be lacking from Section 4 onwards. I don’t quite get the motivation behind the product of graphs and death-time filtrations.
> ```
> We thank the reviewer for raising this comment. On line 454 - 459 of the original submission, we explained what the motivation behind death-time filtrations are. In the construction of the max versions of $\operatorname{RePHINE}^{Spec}, \operatorname{RePHINE}, $ and $\operatorname{EC}$, the edge value $f_e(v, w)$ for an edge $(v, w)$ is defined as the minimum of $f_v(w_1)$ and $f_v(w_2)$. On the other hand, $f_v(w_1)$ and $f_v(w_2)$ are the birth times of the vertices $w_1$ and $w_2$ in the vertex filtration. Thus, a motivating question to ask is - ``what if we replace the birth time in the definition of the max versions with the death time?". Based on your input, we have also moved the discussion on death-time filtrations to the appendix in the current submission.
>
> A key motivation for us to study graph products is that, when dealing with large or complex graphs that have the structural property of being some product, it is often easier to work with the components of the graph product rather than the graph as a whole. This decomposition can be particularly useful in how they relate to persistent homology methods and spectral methods. In particular, we were interested in how much topological information about the whole graph we can recover by just analyzing the components. This is for example the content of Proposition 3, 4, and 5 in the original manuscript. In Proposition 3, we show that the product of vertex filtrations is the max of vertex colors. In Proposition 4, the product of edge filtrations is the natural edge coloring on the product. These two propositions give a natural way to decompose two color-based filtrations of graph products in terms of its components. Finally, Proposition 5 gives a way to compute the 0-dimensional persistence diagram of the graph product based on just its components.
>
> We have also added a description of these motivation for graph products at the beginning of the section.
>
> ```
> Section 5 seems something new, although I am not sure about its applicability as such.
> ```
> We thank the reviewer for raising this comment. Please see our explanation for the motivation of graph products in the response to the previous comment.
>
> ```
> I didn’t quite understand the overall point being made by Section 7.
> ```
>
> We thank the reviewer for raising this question. The overall point of this section is to analyze the persistent homology (PH) and Euler characteristics (EC) of two types of graph representations of a molecule - the Bond Coloring Representation and Bond Counting Representation. The purpose is to analyze under which conditions do the two representations have the same expressivity (ie. Proposition 7 and 8) and when they differ (ie. the remarks near the end of the section). The point we make is that neither subsumes the other, but they are equivalent under specific cases.

---

> ### Author Response · Authors · 2024-11-27
>
> ```
> In Definition 8, what does $h_1 \sim h_2$ mean?
> ```
> We thank the reviewer for raising this question. By $h_1 \sim h_2$, we mean that $h_1$ and $h_2$ are related by an edge in the graph $H$. We have added a clarification of this notation in the definition now.
>
> ```
> Line 491: Bond Coloring Representation -> Bond Counting Representation?
> ```
> We thank the reviewer for point out this typo. We have corrected the typo in the updated manuscript now.
>
> ```
> The stability section just talks about the stability of $RePHINE$ diagrams, which is a straightforward extension of the stability of persistence diagrams. What about the stability of $RePHINE^{spec}$? Can that be characterized as well? What about the stability of EC?
> ```
> Thanks to this question raised by the reviewer (and also reviewer ZQBM), we have looked into the stability of $\operatorname{RePHINE}^{Spec}$ and $\operatorname{EC}$. In the new updated manuscript, we have proven that $\operatorname{RePHINE}^{Spec}$ is locally stable under the assumption that the functions $f_v: X \to \mathbb{R}$ (which assigns each vertex color type with a real value) and $f_e: X \times X/\sim \to \mathbb{R}_{>0}$ (which assigns each edge volor type with a positivie real value) are both injective.
>
> By **locally stable**, we mean that tiny variations in the filtration functions used will not drastically change the output of $\operatorname{RePHINE}^{Spec}$. In our original submission, what we proved for $\operatorname{RePHINE}$ is actually somewhat stronger - we showed that the bottleneck distance of any two RePHINE diagrams on the same graph are bounded in terms of some variant of the $\ell^{\infty}$ metric on the filtration functions that created them. This is a global statement of stability, whereas it suffices for local stability to hold in applications.
>
> While $\operatorname{RePHINE}$ was shown to be globally stable by us, we also showed that $\operatorname{RePHINE}^{Spec}$ is not globally stable. Furthermore, when injectivity does not hold, it may not be locally stable either (but on a measure zero subset of the space of filtration functions). We have provided counter-examples for this at the end of Appendix A.3 (Example 1).
>
> To discuss the word **local**, we needed a notion of topology on $(f_v, f_e)$. The convenient choice for us turned out to be thinking of $(f_v, f_e)$ as an element of $\mathbb{R}^{n} \times \mathbb{R}^m$ (where $n$ is the size of the vertex coloring set $X$ and $m$ is the size of the correspondent edge coloring set). In this case, $f_v$ (resp. $f_e$) being injective is the same as saying its coordinates have no repeated entries. The main idea behind proving the local stability of $\operatorname{RePHINE}^{Spec}$ was that the proof for the stability of $\operatorname{RePHINE}$ may be adapted in a convex open neighborhood of $f_v$ in $\mathbb{R}^n$ and $f_e$ in $\mathbb{R}^m$.
>
> Finally, for EC, We have also found that the authors of [1] have proven the stability for EC already in their Proposition 3.2.
>
> -----
> Many thanks for your insightful review and constructive feedback. We hope your concerns have been satisfactorily addressed, and if so, would appreciate if you could revisit your score to reflect the same. We are also committed to engaging further if you have any additional questions, concerns, or suggestions.
>
>
> [1] Dłotko, Paweł and Gurnari, Davide, **Euler characteristic curves and profiles: a stable shape invariant for big data problems**

---

> > ### Comment · Reviewer_NVLY · 2024-11-30
> >
> > I thank the authors for their efforts in providing an extensive rebuttal. I have read all the answers. However, I still have a few concerns.
> >
> > I don't completely understand the usage of the word "local" in the stability. What sort of locality argument is the statement making? As far as I understand, the only additional condition needed on the filtration functions is that they need to be injective, which in my opinion, is not a very strong requirement. One could always make a filtration function injective, by perturbing the points by epsilon wherever injectivity fails.
> >
> > The motivation for product of graphs is still unclear to me. I get the part that it is better to analyze the graphs component by component than in its entirety. I am still not convinced about the use case for product of graphs. Are there any datasets which have product graphs occurring naturally?
> >
> > The flow still seems a bit lacking from Section 4 onwards.
> >
> > I have revised by score.
> >
> > Fwiw, the revisions made to the paper are substantial, especially in the stability section. I am not sure if the rebuttal time window is enough for going through the new draft.

---

> > > ### Author Response · Authors · 2024-12-01
> > >
> > > Thank you for your follow-up comments and the considerations to raise your scores! We will address your comments and concerns as follows:
> > >
> > > ```
> > > I don't completely understand the usage of the word "local" in the stability. What sort of locality argument is the statement making? As far as I understand, the only additional condition needed on the filtration functions is that they need to be injective, which in my opinion, is not a very strong requirement. One could always make a filtration function injective, by perturbing the points by epsilon wherever injectivity fails.
> > > ```
> > > Thank you for giving us the opportunity to clarify our usage of the terminology ``local stability". In experiments, what we often think of stability is the idea that tiny variations in the parameters should cause tiny changes in the outcome.
> > >
> > > In our statement for the (global) stability of RePHINE in Theorem 4, our precise statement is that - for all filtrations functions $f = (f_v, f_e)$ and $g = (g_v, g_e)$, the distance of their correspondent RePHINE diagrams is bounded by some norm on $f - g$ (more precisely $3||f_e - g_e||_{\infty} + ||f_v - g_v||$). This is global in the sense that this **inequality is valid for any two filtration functions $f$ and $g$ that we choose**. This is also stronger than what we needed for "tiny variations in the parameters should cause tiny changes in the outcome", because that only required the inequality to hold within a neighborhood of $f$.
> > >
> > > What we stated for the ``local stability" of $\operatorname{RePHINE}^{Spec}$ is as follows: Given $f = (f_v, f_e)$ and $g = (g_v, g_e)$ (both injective),  the inequality bounding the distance of their correspondent $\operatorname{RePHINE}^{Spec}$ diagrams **holds only when $g$ is in sufficiently close neighborhood** of $f$. In the actual proof, it turns out the choice of the neighborhood is not that restrictive (any convex open set not containing the points that have repeated entries should do).
> > >
> > > However, for the case of $\operatorname{RePHINE}^{Spec}$, the inequality is not true for any two choice of injective $f$ and $g$'s (ie. it is not globally stable). The injectivity assumption is needed because $\operatorname{RePHINE}^{Spec}$ may not be locally stable at $f = (f_v, f_e)$ if the functions $f_v$ and $f_e$ are not injective. We also gave counter-examples illustrating both of them in Example 1 of Appendix A.3.
> > >
> > > ```
> > > The motivation for product of graphs is still unclear to me. I get the part that it is better to analyze the graphs component by component than in its entirety. I am still not convinced about the use case for product of graphs. Are there any datasets which have product graphs occurring naturally?
> > > ```
> > > We thank the reviewer for raising this question. There are real datasets where graph products find use; e.g., the authors of [1] use  (a) the  COIL-20 dataset for multi-view object clustering, and (b) real air-quality spatiotemporal data with the objective of learning one graph factor for estimating dependencies between air quality measurements in different cities and another graph factor to model the seasonal variations. The common theme is that they show up naturally for multi-view and/or multi-domain data.
> > >
> > > ```
> > > The flow still seems a bit lacking from Section 4 onwards.
> > > ```
> > > We thank the reviewer for raising this comment. Should the paper be accepted, we will add an explanation on how graph products appear naturally for multi-view and/or multi-domain data. Together with the coloring and counting representations for molecular data, the later part of the paper can be viewed coherently as **Datatype-informed topological considerations**. We will also restructure slightly to organize them as such.
> > >
> > > -----
> > > Thank you again for your review and comments! We hope this answers the concerns / questions you have raised in the previous comment, and we would be grateful if you could raise the score to reflect that. Please let us know if you have any further questions or suggestions.
> > >
> > > [1] Sai Kiran Kadambari and Sundeep Prabhakar Chepuri, **Product Graph Learning from Multi-domain Data with Sparsity and Rank Constraints**.

---

> > > ### Author Response · Authors · 2024-12-03
> > >
> > > Dear Reviewer NVLY,
> > >
> > > As the discussion period is ending soon, we wanted to check-in again. We hope that our previous comment has addressed your remaining concerns. If so, we would appreciate it if you can considering raising the score to reflect the same. If you have any other questions/concerns, we are happy to answer them. Thank you again for your feedback and help to improve the paper!

---

### Official Review · Reviewer_qz4P · 2024-11-04

**Soundness:** 3
**Presentation:** 4
**Contribution:** 3
**Rating:** 6
**Confidence:** 3

**Summary:**

This paper provides a rigorous theoretical analysis of different topological descriptors (TDs) used to enhance graph neural networks (GNNs), focusing on Euler characteristics (EC), persistent homology (PH), and Laplacian spectrums. The authors examine these TDs through three key lenses: expressivity (representational power), stability (robustness to perturbations), and computational efficiency. They introduce a novel descriptor called RePHINE_Spec that combines spectral features with RePHINE and prove it is strictly more expressive than existing methods. They also propose a new metric to assess the stability of RePHINE and analyze weaker variants of several descriptors to address computational costs. The work includes theoretical results about filtrations on graph products and applications to molecular graphs.

**Strengths:**

The paper's primary strength lies in its comprehensive theoretical treatment of topological descriptors, providing rigorous proofs and clear characterizations of their properties. The authors make several significant theoretical contributions, including proving the equivalence of EC and max EC diagrams, establishing stability bounds for RePHINE, and characterizing filtrations on graph products. The work is well-structured, building from fundamental concepts to more complex settings. The paper also introduces useful variants of existing methods that maintain expressivity while reducing computational costs.

**Weaknesses:**

The paper is heavily theoretical and could benefit from more empirical validation of the proposed methods, particularly RePHINE_Spec. While the theoretical advantages are well-established, there is limited discussion of practical implementation challenges or computational benchmarks comparing the different methods.

**Questions:**

- How does the computational complexity of RePHINE_Spec compare to existing methods in practice, and what are the memory requirements for storing the additional spectral information?
- Can the authors provide empirical evidence that the improved expressivity of RePHINE_Spec translates to better performance on real-world tasks?
- Could the theoretical framework developed in this paper be extended to handle dynamic or temporal graphs where the topology evolves over time?

---

> ### Author Response · Authors · 2024-11-27
>
> We are grateful to Reviewer qz4P for their time and detailed comments. We reply to your comments/questions below.
>
> ```
> How does the computational complexity of $RePHINE^{Spec}$ compare to existing methods in practice, and what are the memory requirements for storing the additional spectral information?
> ```
>
> We thank the reviewer for raising this question. Compared to RePHINE, $\operatorname{RePHINE}^{Spec}$ needs to store an additional $\gamma$ parameter. While the memory capacity may seem daunting (for example, for a connected tree with 4000 vertices that die at the same time, each $\gamma$ parameter would have a list of 3999 values). In practice, there are various implementation techniques to optimize the memory needed to be used. For example, if multiple vertices die at the same time in the same component, then they really just need to point to the same list. In practice, there is also no need to calculate the entire list of non-zero eigenvalues, often just computing the smallest non-zero eigenvalue (ie. the Fiedler value / algebraic connectivity) can already give a lot of information. There are many optimized methods to do this, such as minimizing the Ralyeigh quotient iteration, etc.
>
> ```
> Can the authors provide empirical evidence that the improved expressivity of $RePHINE^{Spec}$ translates to better performance on real-world tasks?
> ```
> We greatly appreciate your question. We focus here on several key theoretical underpinnings from multiple perspectives to bridge prominent gaps in our understanding of various TDA methods from a machine learning perspective. We fully agree that a comprehensive empirical investigation would be useful, and hope that the theoretical insights presented in this work pave way for such detailed empirical investigations in future studies.
>
> ```
> Could the theoretical framework developed in this paper be extended to handle dynamic or temporal graphs where the topology evolves over time?
> ```
>
> We thank the reviewer for raising this question. The answer we believe for temporal graphs is yes. The main idea is that $\operatorname{RePHINE}^{Spec}$ is a general filtration method that one can consider augmenting any model that uses color-based PH with, and weaker invariants are general filtration methods that one can consider changing any model that uses color-based PH with to address for computational costs. There are existing methods that have applied persistent homology in temporal network analysis [1]. While the authors used a form of ``zigzag persistence", it shows promise that the zigzag persistence may be combined with color-based filtrations to apply to temporal networks.
>
> -----
> Many thanks for your insightful review and constructive feedback. We hope your concerns have been satisfactorily addressed, and if so, would appreciate if you could revisit your score to reflect the same. We are also committed to engaging further if you have any additional questions, concerns, or suggestions.
>
> [1] Audun Myers, David Muñoz, Firas A Khasawneh, and Elizabeth Munch, **Temporal network analysis using zigzag persistence**

---

> ### Author Response · Authors · 2024-12-03
>
> Dear Reviewer qz4P,
>
> As the discussion period is ending soon, we wanted to check-in again. If you have any specifc questions or concerns, we would be happy to address them. Thank you again for your feedback and help to improve the paper.

---

> > ### Comment · Reviewer_qz4P · 2024-12-03
> >
> > Thank you for the response. I have no more questions.

---

### Official Review · Reviewer_ZQBM · 2024-11-04

**Soundness:** 2
**Presentation:** 2
**Contribution:** 2
**Rating:** 6
**Confidence:** 4

**Summary:**

This paper presents new topological descriptors, especially RePHINE and RePHINESpec, designed to enhance the capabilities of GNNs by enhancing how well they capture complex graph features like cycles. By analyzing expressivity, stability, and computational efficiency, the authors compare various descriptors and explore practical applications, including insights for molecular graphs.

**Strengths:**

1. The paper covers a range of topological descriptors and presents a detailed comparison of their expressivity, stability, and computational expenses.
2. The paper considers applications in molecular graphs, offering insights that could benefit chemistry and drug discovery.

**Weaknesses:**

1. The definitions and theoretical explanations are dense and without sufficient explanations. This can be very difficult to follow for people with limited background.
2. The focus is primarily on color-based filtrations. Expanding the discussion to include other kinds of filtrations, would give a more comprehensive perspective.
3. . While the paper presents more expressive descriptors like RePHINESpec, it doesn’t address the computational cost, particularly in large graphs or real-time applications.

**Questions:**

1. The paper’s applications are primarily molecular. Can this be expanded to other fields that use GNNs, such as social network analysis?
2. How does the use of topological descriptors like RePHINESpec compare with traditional GNN methods, particularly in tasks that do not require high expressivity?
3. The paper proposes several filtrations for molecular graphs. What criteria were used to select these?
4. What is the STABILITY OF REPHINE DIAGRAMS? explanation is missing. Can the authors provide practical examples where stability significantly impacts application outcomes?
5. How sensitive are the proposed descriptors (especially RePHINESpec) to variations in parameters such as filtration type or depth?

---

> ### Author Response · Authors · 2024-11-27
>
> We are grateful to Reviewer ZQBM for their time and detailed comments. We reply to your comments/questions below.
>
> ```
> The paper’s applications are primarily molecular. Can this be expanded to other fields that use GNNs, such as social network analysis?
> ```
> We thank the reviewer for raising the question. While our focus is primarily molecular, there are numerous other models that incorporate persistent homology (PH) diagrams that can be easily adapated to our proposed models in the manuscript. One such example of PersLay [1], which is a simple and flexible layer in neural network to process topological data and has been used in application to analyze data from the social media site Reddit. Another example is the application of persistent homology in temporal network analysis [2]. While the authors used a form of ``zigzag persistence", zigzag persistence may be combined with color-based filtrations to apply to temporal networks. The main idea is that $\operatorname{RePHINE}^{Spec}$ is a general filtration method that one can consider augmenting any model that uses color-based PH with.
>
> ```
> How does the use of topological descriptors like RePHINESpec compare with traditional GNN methods, particularly in tasks that do not require high expressivity?
> ```
> We thank the reviewer for raising this question. What filtration methods like $\operatorname{RePHINE}^{Spec}$ and $\operatorname{RePHINE}$ provide are not just expressibity, but also access to global information. Traditional GNN methods have an inherent inductive bias towards local information and can run into data/network homophily, so augmenting global information is what we wanted to emphasize with this work. For example, for any work where you want to count the number of cycles in a graph, the presence of these kind of TDs are still needed.
>
> ```
> The paper proposes several filtrations for molecular graphs. What criteria were used to select these?
> ```
>
> We thank the reviewer for raising this question. We considered two types of filtrations - one based on vertex coloring and other based on edge coloring, and several topological descriptors along these two types of filtrations. Our original goal was to look at how topological descriptors interact with color-based filtrations of graphs, which most commonly includes persistent homology (PH). We noted that RePHINE is a more expressive version of PH, so we also included that as well. There is a classical theorem (see Appendix B for a more precise description) that asserts that zeroth homology is the kernel of graph Laplacian, so a natural generalization for us was to consider the non-zero eigenvalues of the Laplacian. Finally, we also wanted to consider weaker invariants to address for computational costs, hence we considered the Euler characteristics (EC) and the max versions of $\operatorname{RePHINE}^{Spec}, \operatorname{RePHINE},$ and $\operatorname{EC}$.
>
> ```
> What is the STABILITY OF REPHINE DIAGRAMS? explanation is missing. Can the authors provide practical examples where stability significantly impacts application outcomes?
> ```
>
> We thank the reviewer for raising this question. By the ``stability of RePHINE diagram", we mean that RePHINE diagrams are robust to slight perturbations in the input filtrations. This notion is made more precise in the statement of Theorem 4 in the original submission. Specifically, Theorem 4 states that the bottleneck distances (Definition 7) of two RePHINE diagrams on the same graph may be explicitly bounded in terms of the $\ell^{\infty}$-norms of the input functions. We have also added further explanation to this before the statement of Theorem 4 in the edited submission.
>
> There are practical examples where the stability of persistent homology (PH) have benefitted applications outcomes. In the original paper by Cohen-Steiner et al. [3] proving bottleneck stability, the authors applied their stability results to the estimation of the homology of a closed subset of a metric space from a finite point sample, and make concrete statements about the stability of barcodes (a representation of persistence diagrams) with relevance to image processing and morphology.

---

> ### Author Response · Authors · 2024-11-27
>
> ```
> How sensitive are the proposed descriptors (especially RePHINESpec) to variations in parameters such as filtration type or depth?
> ```
>
> We thank the reviewer for raising this question. In our original submission, we already noted that the stability of persistent homology was shown by Cohen-Steiner et al. [3]. Furthermore, we showed in the original submission that $\operatorname{RePHINE}$ is stable under our proposed bottleneck distance, meaning that tiny variations in the filtration functions used will not drastically change the output of $\operatorname{RePHINE}$. What we proved is actually somewhat stronger - we showed that the bottleneck distance of any two RePHINE diagrams on the same graph are bounded in terms of some variant of the $\ell^{\infty}$ metric on the filtration functions that created them. This is a global statement of stability, whereas it suffices for local stability to hold in applications.
>
> Thanks to this question raised by the reviewer (and also reviewer NVLY), we have looked into the stability of $\operatorname{RePHINE}^{Spec}$ and $\operatorname{EC}$. In the new updated manuscript, we have proven that $\operatorname{RePHINE}^{Spec}$ is locally stable under the assumption that the functions $f_v: X \to \mathbb{R}$ (which assigns each vertex color type with a real value) and $f_e: X \times X/\sim \to \mathbb{R}_{>0}$ (which assigns each edge volor type with a positivie real value) are both injective. When injectivity does not hold, we have also provided a counter-example (Appendix A.3, Example 1) to show that local stability fails.
>
> To discuss the word **local**, we needed a notion of topology on $(f_v, f_e)$. The convenient choice for us turned out to be thinking of $(f_v, f_e)$ as an element of $\mathbb{R}^{n} \times \mathbb{R}^m$ (where $n$ is the size of the vertex coloring set $X$ and $m$ is the size of the correspondent edge coloring set). In this case, $f_v$ being injective is the same as saying its coordinates have no repeated entries. The main idea behind proving the local stability of $\operatorname{RePHINE}^{Spec}$ was that the proof for the stability $\operatorname{RePHINE}$ may be adapted in a convex open neighborhood of $f_v$ in $\mathbb{R}^n$ and $f_e$ in $\mathbb{R}^m$.
>
> We have also found that the authors of [4] have proven the stability for EC already in their Proposition 3.2.
>
> -----
> Many thanks for your insightful review and constructive feedback. We hope your concerns have been satisfactorily addressed, and if so, would appreciate if you could revisit your score to reflect the same. We are also committed to engaging further if you have any additional questions, concerns, or suggestions.
>
> [1] Mathieu Carrière, Frédéric Chazal, Yuichi Ike, Théo Lacombe, Martin Royer, and Yuhei Umeda, **PersLay: A Neural Network Layer for Persistence Diagrams and New Graph Topological Signatures**
>
> [2] Audun Myers, David Muñoz, Firas A Khasawneh, and Elizabeth Munch, **Temporal network analysis using zigzag persistence**
>
> [3] David Cohen-Steiner, Herbert Edelsbrunner, and John Harer, **Stability of Persistence Diagrams**
>
> [4] Paweł Dłotko and Davide Gurnari, **Euler characteristic curves and profiles: a stable shape invariant for big data problems**

---

> ### Author Response · Authors · 2024-12-03
> **Follow-up on Rebuttal**
>
> Dear Reviewer ZQBM,
>
> As the discussion period is ending soon, we wanted to check-in again. We have addressed all your questions in our reply. Notably, we expanded on the stability section, included a new discussion on the (local) stability of $\operatorname{RePHINE}^{Spec}$, and found a reference for the stability of EC.
>
> We hope your concerns have been satisfactorily addressed, and if so, would appreciate if you could revisit your score to reflect the same. If you have any specifc questions or concerns, we would be happy to address them. Thank you again for your feedback and help to improve the paper.

---

### Author Response · Authors · 2024-11-27
**Global Comment**

We are grateful to all the reviewers for their time and insightful comments, as well as to the (senior) area, program, and general chairs for their service to the community. We appreciate the reviewers' acknowledgement that the problem we are trying to address - the theoretical pinnings of how topological methods relate to graph neural networks - are important.

We have submitted a new version of our manuscript to address the common questions/concerns raised by the reviewer. Most of the notable changes are colored in orange. Here we summarize the changes we made to address the questions/concerns:

##  Extending Stability
Some reviewers were curious in whether the discussion of stability of RePHINE and persistent homology (PH) can be extended to the other two filtration methods we considered in this manuscript - $\operatorname{RePHINE}^{Spec}$ and $EC$. We first found out that the stability of $EC$ was already shown by Dłotko an Gurnari in [1] (Proposition 3.2).

We then investigated the stability of $\operatorname{RePHINE}^{Spec}$. Specifically, we prove in the new manuscript that $\operatorname{RePHINE}^{Spec}$ is locally stable under the assumption that the functions $f_v: X \to \mathbb{R}$ (which assigns each vertex color type with a real value) and $f_e: X \times X/\sim \to \mathbb{R}_{>0}$ (which assigns each edge volor type with a positivie real value) are both injective.

By **locally stable**, we mean that tiny variations in the filtration functions used will not drastically change the output of $\operatorname{RePHINE}^{Spec}$. In our original submission, what we proved for $\operatorname{RePHINE}$ is actually somewhat stronger - we showed that the bottleneck distance of any two RePHINE diagrams on the same graph are bounded in terms of some variant of the $\ell^{\infty}$ metric on the filtration functions that created them. This is a global statement of stability, whereas it suffices for local stability to hold in applications.

While $\operatorname{RePHINE}$ was shown to be globally stable by us, we also showed that $\operatorname{RePHINE}^{Spec}$ is not globally stable. Furthermore, when injectivity does not hold, it may not be locally stable either (but on a measure zero subset of the space of filtration functions). We have provided counter-examples for this at the end of Appendix A.3 (Example 1).

To discuss the word **local**, we needed a notion of topology on $(f_v, f_e)$. The convenient choice for us turned out to be thinking of $(f_v, f_e)$ as an element of $\mathbb{R}^{n} \times \mathbb{R}^m$ (where $n$ is the size of the vertex coloring set $X$ and $m$ is the size of the correspondent edge coloring set). In this case, $f_v$ being injective is the same as saying its coordinates have no repeated entries. The main idea behind proving the local stability of $\operatorname{RePHINE}^{Spec}$ was that the proof for the stability $\operatorname{RePHINE}$ may be adapted in a convex open neighborhood of $f_v$ in $\mathbb{R}^n$ and $f_e$ in $\mathbb{R}^m$.

The discussions on the new results are now in Section 4 of the manuscript, and the correspondent proofs are in Appendix A.3.

## Graph Product Motivation
Some reviewers raised some concerns that the motivations behind the discussion of graph products were not explained clearly. We have added a new explanation at the beginning of the correspondent section. Specifically, a key motivation for us to study graph products is that, when dealing with large or complex graphs that have the structural property of being some product, it is often easier to work with the components of the graph product rather than the graph as a whole. This decomposition can be particularly useful in how they relate to persistent homology methods and spectral methods. In particular, we were interested in how much topological information about the whole graph we can recover by just analyzing the components.

## Static vs Persistent
There was a question raised about whether the topological methods we are considering in our work are **static** or **persistent**. To clarify, for us, a **persistent** topological method is a method that involves examining at how the topological invariants change over time with respect to the color-based filtrations of the graph. A **static** topological method is a method that does not take into account of this change over time. Therefore, all methods we defined in Section 2 of the manuscript are **persistent** topological methods. The reason why we wanted to **persistent** methods over **static** methods is because static methods only give a snapshot of the whole timeline of changes persistent methods can give, so persistent methods are generally more expressive than static methods of the same kind.

## Death Time Filtration
We have moved the discussion of the death time filtration to the appendix.

[1] Paweł Dłotko and Davide Gurnari, **Euler characteristic curves and profiles: a stable shape invariant for big data problems**

---

### Meta-Review · Area_Chair_UgtS · 2024-12-20

**Metareview:**

This submission provides a characterisation of different topological descriptors along the axes of expressivity (that is, their power to _represent_ objects, stability (that is, how robust they are with respect to perturbations), and computation (that is, how fast they are and how easy they are to implement). Moreover, a new descriptor, RePHINE$^\text{Spec}$, is developed, which incorporates information RePHINE—an existing descriptor—and the graph Laplacian, in order to outperform existing descriptors in terms of expressivity. The submission then discusses the aforementioned aspects for the different descriptors, providing more insights into their underpinnings. This, along with the comprehensive scope of the submission, is also one of the main strengths of the submission. I also find the collection of theoretical results to be useful for practitioners with a background in computational topology.

These strengths are, however, marred by certain weaknesses, not all of which could be easily addressed during the rebuttal. Specifically, the following concerns remain largely unaddressed:

1. The overall "narrative" of the paper is lacking—while I and the reviewers understand the relevance of assessing topological descriptors from multiple perspectives, some of the sections are not sufficiently motivated (at least not on first glance). This makes the paper appear to lack some cohesion.
2. In addition, due to the focus on three different "axes" as pointed out above, some of the results are not described in-depth, while other results could be considered as "well-known" already, at least in the computational topology community.
3. Clearly, making such results available to a wider audience is a valid enterprise, and would indeed be the basis for a strong submission, but the paper, in its current form, lack accessibility by non-expert readers, thus impeding the stated goal of the manuscript, namely the dissemination of expressivity results of topological descriptors to a wider machine-learning community.
4. In addition, for such an audience, _some_ empirical components would be highly appreciated, in particular in light of the fact that topological methods may yet have a pivotal role to play in areas like graph machine learning, for instance.

It is for these reasons that I have to suggest rejecting the paper for now. I understand that this is not the desired outcome for the authors, and, as I shall briefly explain below, some of the reviewers could have been more active during the rebuttal. I took all these aspects into consideration, but nevertheless, I believe that this work—which has lots of potential—requires a major round of additional changes before being ready for publication. I see multiple paths going forward, and I hope the authors find one of them appropriate:

1. By improving the accessibility for a general machine-learning audience, together with the addition of empirical results, the authors could easily target another machine-learning conference with their revision.
2. By focusing more on the theoretical aspects of the works, the authors could target a journal like [JACT](https://link.springer.com/journal/41468), which has an audience with a strong background in (computational) topology.
3. Finally, the authors could also target JMLR or a related venue and rewrite their paper as a general _survey_ on such results; notice that a survey would not preclude theoretical results on their own, of course. In this case, a more comprehensive analysis of other descriptors would be warranted.

Regardless of the path taken by the authors, I believe in the potential of the work, and I would encourage them to incorporate some of the salient points raised by reviewers.

**Additional Comments On Reviewer Discussion:**

Reviewer `ZQBM` raised issues about the clarity and accessibility of the work; they also had some questions about technical and proof details. Giving an initially slightly-favourable opinion on the paper, the reviewer unfortunately did _not_ engage further with the authors, leaving it to me to judge to what extent everything has been addressed. I found the provided changes by the authors only partially sufficient, though, and the main weakness raised by the reviewer, concerning the overall accessibility of the work, was unfortunately _not_ sufficiently addressed during the rebuttal.

Reviewer `qz4P` raised issues about the clarity and accessibility of the work as well, wishing for a more empirical assessment of performance in the form of experiments, for instance. Their background indicates familiarity with TDA methods, making the accessibility issues something to heed. The authors addressed some missing explanations for this reviewer, but did not provide a sufficient revision towards improving the accessibility of the work.

Reviewer `NVLY` pointed out issues with the overall narrative or "flow." The reviewer has a strong background in computational topology and appreciated the changes of the authors, in particular with respect to contextualising some of the stability results. However, the reviewer also pointed out that "Fwiw, the revisions made to the paper are substantial, especially in the stability section.". This (see review above) is a sentiment that I share, and, based on the discussion thus far, I would believe that another round of reviews would be warranted.

Reviewer `pi7v` raised concerns about the novelty of the work, citing that while they would appreciate a rigorous study, they feel that the paper is currently not achieving its states objectives very well. While the authors did address several questions, including some clarifications on static and dynamics descriptors, the main issue about the contributions _per se_ could not be addressed in the time window allotted for the rebuttal. It is also unfortunate to see a lack of engagement by reviewer `pi7v`, who failed to acknowledge the rebuttal by the authors. Another round of discussions would have helped and would have provided additional guidance to the authors.

Finally, reviewer `icUB` took umbrage with the broad scope of the paper and, in a similar fashion to `pi7v`, therefore found some contributions incremental. The reviewer engaged with authors in discussions but ultimately concluded that a major revision of the work would be apt.

I base my final verdict on all the shared concerns by the reviewers as well as my own reading of the paper. As mentioned above, I find it extremely relevant to ensure accessibility for a wider audience, as I believe that this will ensure that the work gets the attention it deserves.

As I summarise this rebuttal phase, I am taken aback by some of the non-responding reviewers; this might indicate a general problem with the setting as it is, in which reviewers often have to respond very quickly to multiple such queries. It is for a consequence of these time pressures on reviewers that the authors might have the impression that their work was not reviewed by experts. However, I can assure them that, among the pool of reviewers who submitted their reviews here, many of them have a strong background in computational topology, while all of them have a background in machine learning. As such, these reviewers are qualified to judge the merits of the work, and the authors are encouraged to take their concerns and misapprehensions into account.

I want to conclude by stating that I believe this has the potential to be a highly-impactful submission, and I am certain that the work will find its appropriate platform in the future.

---

### Decision · Program_Chairs · 2025-01-22

Reject